# Epithelial Transport in Disease: An Overview of Pathophysiology and Treatment

**DOI:** 10.3390/cells12202455

**Published:** 2023-10-15

**Authors:** Vicente Javier Clemente-Suárez, Alexandra Martín-Rodríguez, Laura Redondo-Flórez, Carlota Valeria Villanueva-Tobaldo, Rodrigo Yáñez-Sepúlveda, José Francisco Tornero-Aguilera

**Affiliations:** 1Faculty of Sports Sciences, Universidad Europea de Madrid, Tajo Street, s/n, 28670 Madrid, Spain; vctxente@yahoo.es; 2Group de Investigación en Cultura, Educación y Sociedad, Universidad de la Costa, Barranquilla 080002, Colombia; 3Department of Health Sciences, Faculty of Biomedical and Health Sciences, Universidad Europea de Madrid, C/Tajo s/n, Villaviciosa de Odón, 28670 Madrid, Spain; lauraredondo_1@hotmail.com (L.R.-F.); c.vtobaldo@gmail.com (C.V.V.-T.); 4Faculty of Education and Social Sciences, Universidad Andres Bello, Viña del Mar 2520000, Chile; rodrigo.yanez.s@unab.cl

**Keywords:** epithelial transport, ion transport, hormonal regulation, genetic disorders, pathophysiology, renal disease

## Abstract

Epithelial transport is a multifaceted process crucial for maintaining normal physiological functions in the human body. This comprehensive review delves into the pathophysiological mechanisms underlying epithelial transport and its significance in disease pathogenesis. Beginning with an introduction to epithelial transport, it covers various forms, including ion, water, and nutrient transfer, followed by an exploration of the processes governing ion transport and hormonal regulation. The review then addresses genetic disorders, like cystic fibrosis and Bartter syndrome, that affect epithelial transport. Furthermore, it investigates the involvement of epithelial transport in the pathophysiology of conditions such as diarrhea, hypertension, and edema. Finally, the review analyzes the impact of renal disease on epithelial transport and highlights the potential for future research to uncover novel therapeutic interventions for conditions like cystic fibrosis, hypertension, and renal failure.

## 1. Introduction

Epithelial transport is a critical physiological process crucial for maintaining the proper functioning of various organs within the human body. Epithelial cells are distributed across diverse tissues and organs like the lungs, kidneys, and intestines, responsible for selectively transporting ions, nutrients, and other substances across cell membranes [1,2]. This intricate transport system is essential for maintaining fluid and electrolyte balance, nutrient absorption, waste removal, and pH regulation. Disruptions in these mechanisms can lead to various pathophysiological conditions, including cystic fibrosis, diarrhea, and hypertension. In the case of cystic fibrosis, a genetic defect in the cystic fibrosis transmembrane conductance regulator (CFTR) protein impairs chloride transport across epithelial cell membranes, resulting in the accumulation of thick mucus in the lungs and other organs, leading to chronic respiratory infections and other complications [3].

Similarly, disruptions in intestinal epithelial transport can cause diarrhea, characterized by excessive fluid loss and dehydration. This can occur due to factors such as bacterial or viral infections, inflammatory bowel disease, and certain medications [4]. Worldwide, viruses are the primary causative agents for acute gastroenteritis, constituting the majority of diagnosed cases of community-acquired diarrhea [5]. In the case of hypertension, disruptions in epithelial transport within renal tubules can lead to impaired sodium and water balance, resulting in elevated blood pressure [6].

Epithelial transport, involving the movement of ions, nutrients, and substances across cell membranes, is vital for the proper function of organs like the lungs, kidneys, and intestines. For instance, in the lungs, epithelial cells facilitate oxygen and carbon dioxide transport across the alveolar membrane, crucial for gas exchange and oxygen de-livery to body tissues. Similarly, in the kidneys, renal tubular epithelial cells selectively reabsorb and excrete ions and substances, maintaining fluid and electrolyte balance [7,8].

Recent advances in our understanding of epithelial transport have elucidated the intricate mechanisms governing this process. Research has unveiled the essential role of ion channels, transporters, and pumps in selectively moving ions and substances across cell membranes. Furthermore, various signaling pathways and regulatory mechanisms, such as phosphorylation, ubiquitination, and membrane trafficking, have been identified to control the activity of these transport proteins [9,10]. Recent studies have also underscored the significance of the gut microbiota in regulating epithelial transport within the intestine. The gut microbiota, comprising trillions of microorganisms residing in the gastrointestinal tract, can influence intestinal epithelial cell function through mechanisms like metabolite production, neurotransmitters, and signaling molecules, impacting epithelial transport and other aspects of gut physiology, including immune function and gut barrier integrity [11]. Additionally, recent research has explored the potential of targeting epithelial transport mechanisms for treating various disorders. For example, drugs modulating ion channels and transporters, including diuretics, have been employed in treating conditions such as hypertension and heart failure [12]. Other approaches like gene therapy and small-molecule inhibitors are in development to target specific transport proteins and correct epithelial transport defects in disorders like cystic fibrosis [13]. The critical role of epithelial transport in human physiology and its involvement in the pathophysiology of various disorders emphasize the need for ongoing research into these underlying mechanisms. Hence, this narrative review aims to provide a comprehensive exploration of the pathophysiological mechanisms governing epithelial transport and its significance in the development of various illnesses. Advances in our comprehension of epithelial transport mechanisms and their regulation have the potential to lead to novel therapies for a range of conditions, including cystic fibrosis, hypertension, and other disorders (see Figure 1).

## 2. Methodology

In this study, a comprehensive literature search was performed using primary and secondary sources, such as scientific articles, bibliographic indexes, and databases including PubMed, Scopus, Embase, Science Direct, Sports Discuss, ResearchGate, and the Web of Science. The search utilized MeSH-compliant keywords related to epithelial and ion, water, nutrient, hormonal regulation, and transport, as well as the pathophysiology of diarrhea, hypertension, edema, and renal disease. The search spanned from the 1990s to 1 January 2023. From a total of 45,272 based on the initial inquiry, the following exclusion criteria were applied: (i) studies with inappropriate or not relevant topics, which were not pertinent to the main focus of the review, and (ii). Ph.D. dissertations, conference proceedings, and unpublished studies. We included all of the articles that met the scientific methodological standards and had implications for any of the subsections in which this article is distributed: types of epithelial transport, ion transport, and nutrient transport; hormonal regulation of epithelial transport; genetic disorders of epithelial transport; and the pathophysiology of diarrhea, hypertension, edema, and renal disease. The information treatment was carried out by a collective of five authors who participated in the review process. A consensus was achieved within the group on the selection of these specific subtopics. Subsequently, the application of exclusion criteria was carried out on studies that employed obsolete data, addressed inappropriate subjects, or were not written in the English language. The five authors engaged in a discussion on the articles, ultimately selecting 222 papers that were deemed relevant to the search parameters pertinent to our study’s purpose. The authors responsible for conducting the study selection also undertook the task of extracting information from the chosen studies. The selection of research was conducted in an impartial manner, and subsequent discussions of the findings led to the development of the current narrative review.

## 3. Types of Epithelial Transport

Epithelial transport is the process through which substances are selectively transported across epithelial cells to maintain homeostasis in the body. There are three primary mechanisms of epithelial transport: transcellular, paracellular, and vesicular transport.

### 3.1. Transcellular Transport

Transcellular transport involves the selective movement of substances across epithelial cells via transmembrane proteins, such as ion channels, transporters, and pumps. These proteins regulate the movement of ions and other solutes through the cell membrane, allowing precise control over the composition of body fluids [14].

Epithelial cells form the barrier between the external environment and the internal environment of the body. The different types of epithelial transport are important for the maintenance of the body’s internal environment, including the regulation of pH, osmolarity, and ion concentrations. Transcellular transport is critical for the movement of substances across the epithelial barrier, including the transport of nutrients, water, and electrolytes. For example, the transcellular movement of sodium and chloride ions across the epithelial barrier is a critical step in the generation of the transepithelial potential difference, which is essential for the absorption of nutrients in the intestines [10].

### 3.2. Paracellular Transport

Paracellular transport occurs between cells and involves the passage of substances through the tight junctions between epithelial cells. These tight junctions prevent the unregulated movement of substances between cells but can be selectively permeable to allow the passage of certain molecules [15,16]. Paracellular transport plays a role in regulating the movement of water and solutes across the epithelial barrier. Tight junctions between epithelial cells create a selectively permeable barrier that allows the regulated movement of certain molecules between cells while preventing the uncontrolled passage of substances. The regulation of paracellular transport is crucial for the maintenance of epithelial barrier function, as alterations in this mechanism can lead to the disruption of barrier function and the development of diseases such as inflammatory bowel disease [11]. Paracellular transport is a type of epithelial transport that plays a critical role in regulating the movement of water and solutes across the epithelial barrier. Unlike transcellular transport, which involves the movement of molecules through epithelial cells, paracellular transport involves the movement of molecules between epithelial cells through tight junctions. The tight junctions that form between epithelial cells create a barrier that prevents the unregulated movement of molecules across the epithelium. However, certain ions and small molecules can pass through the tight junctions via selective channels, which are regulated via specific proteins, such as claudins and occludins [17,18]. The movement of these molecules is tightly regulated and can be influenced by various factors, such as pH, osmotic pressure, and the concentration of ions and solutes on either side of the epithelium. Disruptions in paracellular transport can lead to various diseases, such as inflammatory bowel disease (IBD) and celiac disease. In IBD, alterations in the tight junctions of the intestinal epithelium led to increased permeability, which allows harmful substances to pass through the epithelium and trigger an immune response [19]. In celiac disease, gluten triggers the release of zonulin, a protein that regulates tight junctions, leading to increased permeability and the development of inflammation [20].

### 3.3. Vesicular Transport

Vesicular transport involves the movement of substances through vesicles, which are small membrane-bound compartments that bud off from the plasma membrane. This process can be either endocytosis, in which substances are taken up into the cell, or exocytosis, in which substances are released from the cell [21]. These different mechanisms of epithelial transport play crucial roles in maintaining homeostasis in the body. For example, transcellular transport is responsible for the movement of nutrients, electrolytes, and other solutes across the epithelial barrier, while paracellular transport helps to maintain the integrity of the epithelial barrier and regulate the movement of water and solutes. Vesicular transport is involved in the uptake and secretion of substances, including hormones and neurotransmitters. Additionally, the use of mammalian epithelial cell lines in experimental studies has provided valuable insights into the biosynthetic and recycling processes of apical and basolateral plasma-membrane proteins. Furthermore, these experiments have successfully identified the specific components responsible for guiding the movement of apical and basolateral proteins along these channels. The aforementioned components encompass apical and basolateral sorting signals, i.e., adaptors facilitating basolateral signals, as well as docking and fusion proteins involved in vesicular trafficking. In contemporary investigations utilizing live-cell imaging techniques, researchers have been able to see and analyze sorting processes occurring within epithelial cells in real time. These studies have shed light on the significant contributions of actin, microtubules, and motors in the intricate organization of post-Golgi trafficking [22].

Vesicular transport is important for the uptake and release of substances into and out of cells. For example, the release of hormones and neurotransmitters from endocrine and nervous system cells involves vesicular transport. Vesicular transport can also be involved in the uptake of nutrients and other substances, such as cholesterol and iron, into cells [12]. Vesicular transport is critical for processes such as cell signaling, protein secretion, and membrane recycling. The importance of vesicular transport is illustrated by studies showing that disruptions in this mechanism can lead to various diseases. For example, defects in vesicular transport have been implicated in neurological disorders such as Alzheimer’s disease and Parkinson’s disease [23,24]. Vesicular transport has also been shown to be involved in the pathogenesis of infectious diseases, such as viral infections, by facilitating the entry of pathogens into cells [25]. The regulation of vesicular transport is complex and involves multiple proteins and signaling pathways. For example, the Rab family of small GTPases plays a critical role in the regulation of vesicular trafficking [26]. Additionally, vesicular transport is regulated through various signaling pathways, including the phosphatidylinositol-3-kinase (PI3K) signaling pathway, which is important for the regulation of endocytosis and membrane recycling [27]. 

## 4. Methods for Measuring and Analyzing

### 4.1. Methods for Measuring Mitochondrial Transfer In Vitro and In Vivo

The mitochondria has the capability to regulate the levels of reactive oxygen species (ROS) within the enterocytes, hence playing a crucial role in maintaining epithelial homeostasis through the modulation of apoptosis [28,29]. Mitochondrial transfer is a critical cellular mechanism that allows cells to maintain healthy mitochondria and sustain proper cellular function. Measuring mitochondrial transfer in vitro and in vivo is essential to understand the underlying mechanisms of this process and its potential implications in health and disease [30]. 

In vitro methods provide a controlled and reproducible environment to study mitochondrial transfer. One of the advantages of using fluorescent dyes is that they allow real-time monitoring of mitochondrial transfer from donor to recipient cells. For instance, the MitoTracker dyes are frequently used for staining mitochondria in donor cells, which allows the tracking of the transfer of mitochondria to recipient cells via flow cytometry or confocal microscopy [30,31]. The labeling of mitochondria with fluorescent dyes also provides a reliable and quantitative method to measure the efficiency of mitochondrial transfer between different cell types.

In vitro studies are performed in a controlled environment outside of the living organism, usually using cultured cells. One of the most common methods for measuring mitochondrial transfer in vitro is using fluorescent dyes [32]. The dyes are loaded into donor cells, which are then co-cultured with recipient cells. The transfer of mitochondria can be visualized through fluorescence microscopy, and the amount of transfer can be quantified using flow cytometry [25]. Another in vitro method for measuring mitochondrial transfer is the use of mitochondrial DNA (mtDNA) sequencing. By sequencing mtDNA from donor and recipient cells, the presence of donor mtDNA in recipient cells can be identified, indicating the transfer of mitochondria [33]. The mtDNA sequencing method has the advantage of being able to detect the transfer of functional mitochondria, which can result in improved cellular metabolism and bioenergetics.

However, there are limitations to in vitro methods for measuring mitochondrial transfer. In vitro systems may not fully represent the complexity of in vivo environments, and the experimental conditions may not accurately reflect the physiological and metabolic state of cells in vivo [34]. Additionally, the use of fluorescent dyes and mtDNA sequencing methods may require invasive procedures to load donor cells, which could affect cell viability and induce cellular stress responses [35].

In contrast, in vivo experiments are carried out within a living body. The measurement of mitochondrial transfer in vivo presents more challenges compared to in vitro experiments, mostly attributed to the intricate nature of the organism’s surrounding environment. The utilization of in vivo techniques to measure mitochondrial transfer offers a research framework that is more representative of physiological conditions, hence enhancing the study of intercellular mitochondrial transfer. A frequently employed methodology involves the utilization of animal models, specifically mice or rats, for the purpose of examining the intercellular transport of mitochondria across various tissues or organs. One instance of investigation is the examination of the transfer of mitochondria from mesenchymal stem cells to lung tissue, which has been conducted in mice models of acute lung damage [33].

One in vivo method for measuring mitochondrial transfer is using mitochondrial labeling. Labeling mitochondria with a fluorescent protein, such as green fluorescent protein (GFP), allows the tracking of the mitochondria’s movement and localization within the organism (Hayakawa et al., 2016) [36]. Another in vivo method for measuring mitochondrial transfer is the use of mtDNA sequencing. By sequencing mtDNA from tissues or organs of interest, the presence of donor mtDNA can be identified, indicating the transfer of mitochondria between cells (Islam et al., 2012) [37]. Another in vivo approach for measuring mitochondrial transfer is using mitochondrial DNA (mtDNA) haplotypes. mtDNA haplotypes can be used to track the transfer of mitochondria from maternal sources, which can be particularly useful in the study of maternal–fetal mitochondrial transfer during pregnancy [38]. This approach can provide insight into the role of maternal–fetal mitochondrial transfer in fetal development and disease.

In addition to animal models, advances in imaging techniques have also enabled the visualization of mitochondrial transfer in vivo. For example, two-photon microscopy can be used to track the transfer of mitochondria between cells in live animals [39]. This technique allows the visualization of mitochondrial dynamics and transfer between cells in real time, which can provide valuable information about the mechanisms underlying mitochondrial transfer in vivo.

Every technique used to quantify mitochondrial transfer possesses unique advantages and disadvantages. The utilization of in vitro techniques enables researchers to maintain exact control over the experimental parameters, facilitating accurate observations. Additionally, the application of fluorescent dyes offers a direct approach to visualize and quantify the process of mitochondrial translocation. Nevertheless, these methodologies fail to adequately capture the intricacies of in vivo settings. Performing in vivo procedures presents greater challenges; however, it offers a more authentic portrayal of mitochondrial translocation within living animals. The technique of mitochondrial labeling enables the monitoring and tracing of mitochondria inside an organism, whereas the sequencing of mitochondrial DNA (mtDNA) offers conclusive proof of the transfer of mitochondria. Nevertheless, it is important to note that these methodologies may not effectively capture the precise translocation of mitochondria in particular cell types or organs that are of significance.

The measurement of mitochondrial transfer is of utmost importance in comprehending the fundamental mechanisms governing this phenomenon and its ramifications in the context of both physiological well-being and pathological conditions. Fluorescent dyes and mitochondrial DNA (mtDNA) sequencing are commonly employed in vitro techniques that offer a direct and accurate means of quantifying mitochondrial translocation. In vivo techniques, such as the utilization of mitochondrial labeling and mtDNA sequencing, offer a more accurate depiction of mitochondrial transfer inside living organisms. Every methodology possesses its own set of advantages and drawbacks, and the choice of methodology should be made in accordance with the research inquiry and experimental framework.

### 4.2. Methods to Analyze Epithelial Transport

One of the primary obstacles encountered in the field of epithelial transport research involves the establishment of a model system that effectively replicates the crucial characteristics of epithelial cells [40]. Epithelial transport research commonly employs many in vitro cell culture models, including Caco-2, T-84, and HT-29-Cl.19A cells. The utilization of these cell lines exemplifies a reductionist methodology in the modeling of the epithelium, and they have been extensively employed in many mechanistic investigations, particularly in the exploration of interactions between the epithelium and microorganisms. Nevertheless, cell monolayers fail to effectively represent the interactions between cells and the complex milieu seen in living organisms. The use of three-dimensional (3D) cell cultures has shown considerable potential as a viable approach for investigating drug permeability. Anabazhagan et al.’s work demonstrates the utility of utilizing three-dimensional (3D) Caco-2 cell models for investigating the functionality of epithelial transporters [40]. In addition, it is crucial to supplement investigations conducted on Caco-2 cells with other models in order to eliminate any biases associated with cell-specific responses and consider the intricate nature of the native intestinal environment. Various techniques have been previously employed to evaluate the efficacy of transporters, including everted sac and uptake assays conducted on isolated epithelial cells or isolated plasma membrane vesicles. In light of the difficulties encountered in the domain pertaining to models and the quantification of transport function, this study presents a methodology for cultivating Caco-2 cells in a three-dimensional configuration. Additionally, it outlines the utilization of a Ussing chamber as a viable technique for evaluating serotonin transport, particularly in intact polarized intestinal epithelial tissues [40].

Of course, the kinetics of epithelial membrane transporters play a significant role in the movement of solutes into and out of cells. Mechanistic models are employed for the purpose of examining the transportation of solutes at various scales, including the organ, tissue, cell, or membrane level. King et al. [41] provide a comprehensive analysis of the latest developments in employing computer models for the examination of epithelial transport kinetics on the cellular membrane. Various methods have been employed to create transport phenomenon models that deal with the movement of solutes across the epithelial cell membrane. It is worth noting that a significant number of models included lumped parameters, such as the Michaelis–Menten kinetics, in order to simplify the reaction term associated with transporter-mediated processes. Regrettably, this assumption fails to consider the presence of transporter numbers or the potential influence of external stimuli on membrane transport. In contrast, contemporary mechanistic models of transporter kinetics incorporate the consideration of the number of transporters involved. Researchers can explore the ramifications of physical or chemical disruptions on the system by developing models that closely resemble reality. It is apparent that there is a necessity to enhance the intricacy of mechanistic models that examine the movement of solutes over a membrane. This is crucial in order to acquire a deeper understanding of the interactions between transporters and solutes [41]. 

## 5. Ion Transport

Ion transport plays a crucial role in the normal functioning of different tissues in the body, including the respiratory, digestive, and urinary tracts. Abnormalities in ion transport can lead to various diseases, such as cystic fibrosis and renal tubular acidosis [42]. In the respiratory system, the airway epithelium is responsible for regulating the movement of ions such as chloride, sodium, and bicarbonate. The transport of these ions helps to maintain proper airway hydration, mucociliary clearance, and pH balance. In cystic fibrosis (CF), a genetic disease caused by mutations in the CFTR gene, there is a defect in the chloride channel, leading to reduced chloride secretion and increased sodium absorption, resulting in thick and sticky mucus in the airways, thus leading to chronic lung infections and respiratory failure [22].

In the digestive system, the transport of ions is essential for the secretion and absorption of fluids and electrolytes in the intestine. The small intestine and colon are responsible for the absorption of nutrients and water, while the colon also plays a role in maintaining electrolyte balance. In diarrhea, there is an increase in chloride secretion and a decrease in sodium absorption, leading to excessive fluid loss. In contrast, in constipation, there is reduced secretion of fluids and electrolytes, leading to fecal impaction [43]. 

In the urinary system, the transport of ions such as sodium, chloride, potassium, and bicarbonate is essential for the maintenance of proper electrolyte balance and acid–base homeostasis. The kidney tubules, including the proximal tubule, loop of Henle, distal tubule, and collecting ducts, are responsible for regulating ion transport. Renal tubular acidosis (RTA) is a disease characterized by a defect in the renal tubules, leading to reduced acid secretion and impaired bicarbonate reabsorption, resulting in metabolic acidosis [44].

Further, the pathophysiology of ion transport in various tissues involves the function of ion channels and transporters. These proteins are responsible for the movement of ions across the cell membrane and play a crucial role in regulating the fluid and electrolyte balance in the body [45].

In the urinary system, the transport of ions is regulated via various transporters and channels in the kidney tubules. For example, the sodium–potassium pump (Na+/K+ ATPase) is responsible for the movement of sodium ions out of the cell in exchange for potassium ions. The chloride–bicarbonate exchanger (AE1) is involved in the exchange in bicarbonate ions for chloride ions in the renal tubules. In RTA, mutations in the genes encoding these transporters can lead to defects in acid secretion and bicarbonate reabsorption, leading to metabolic acidosis [46].

In this line, cystic fibrosis (CF) is a genetic disorder that affects multiple organs, including the respiratory and digestive systems [47,48]. CF is caused by mutations in the cystic fibrosis transmembrane conductance regulator (CFTR) gene, which codes for a chloride channel found in various tissues, including the airway epithelial cells, pancreatic ducts, and sweat glands [49]. The lack of pancreatic enzymes leads to the malabsorption of nutrients, causing malnutrition and growth failure in CF patients [47].

Renal tubular acidosis (RTA) is a group of disorders characterized by a defect in the secretion or reabsorption of bicarbonate ions in the renal tubules, leading to metabolic acidosis. The acidosis can cause various symptoms, including weakness, fatigue, confusion, and bone demineralization [50]. There are three types of RTA: distal RTA, proximal RTA, and hyperkalemic RTA. In distal RTA, there is a defect in the secretion of acid in the distal renal tubules, leading to a reduced ability to excrete acid and reabsorb bicarbonate ions. This results in a net loss of bicarbonate ions and metabolic acidosis. Proximal RTA is caused by a defect in the reabsorption of bicarbonate ions in the proximal renal tubules, leading to a net loss of bicarbonate ions and metabolic acidosis. Hyperkalemic RTA is caused by a defect in the secretion of potassium ions in the distal renal tubules, leading to hyperkalemia (high levels of potassium in the blood) and metabolic acidosis [30].

## 6. Water Transport

Water transport is essential for maintaining the body’s fluid balance and normal physiological functions. The movement of water across cell membranes is primarily driven by osmosis, which is the movement of water from an area of low solute concentration to an area of high solute concentration. In different tissues, water transport is regulated via various channels and transporters present on the cell membranes (Verkman et al., 1996) [31]. In the renal tubules, the transport of water occurs in the nephrons, which are the functional units of the kidneys. The movement of water in the nephrons is regulated via the hormone antidiuretic hormone (ADH), which is secreted by the pituitary gland. ADH binds to the V2 receptor present on the basolateral membrane of the epithelial cells in the collecting ducts and increases the expression of the aquaporin-2 channel on the apical membrane. This channel allows the movement of water from the lumen of the collecting ducts to the interstitial fluid and then to the bloodstream. Disruption in ADH secretion or expression of aquaporin-2 channel can lead to conditions like diabetes insipidus, where excessive water loss occurs [32].

In the lungs, water transport occurs across the alveolar–capillary membrane. The movement of water in the lungs is regulated via the oncotic pressure gradient, which is the difference in the concentration of proteins between the plasma and interstitial fluid. The plasma proteins create an osmotic gradient that pulls water from the interstitial fluid to the plasma. Disruption in the oncotic pressure gradient, such as in conditions like hypoalbuminemia, can lead to the accumulation of fluid in the lungs, leading to pulmonary edema [33]. In the gastrointestinal tract, water transport occurs in the small and large intestines. The movement of water in the intestines is regulated via the osmotic gradient created by the solutes present in the luminal contents. The absorption of water occurs in the small intestine, while the large intestine reabsorbs the remaining water and concentrates the fecal matter. Disruption in the absorption or reabsorption of water can lead to conditions like diarrhea or constipation [34].

Disruptions in water transport can lead to various conditions, including dehydration and pulmonary edema. Dehydration occurs when the body loses more water than it takes in, leading to a decrease in total body water volume. This can occur in conditions like excessive sweating, diarrhea, or vomiting. On the other hand, pulmonary edema occurs when there is an accumulation of fluid in the lungs due to disruption in the normal mechanisms of water transport. This can occur in conditions like heart failure or acute respiratory distress syndrome (ARDS) [35].

In the renal tubules, water transport occurs in the nephrons, which are responsible for filtering blood and producing urine [38]. The movement of water in the nephrons is regulated via the hormone antidiuretic hormone (ADH), which is secreted by the pituitary gland. ADH acts on the collecting ducts in the nephrons and increases the expression of the aquaporin-2 (AQP2) channel on the apical membrane of the epithelial cells [39]. AQP2 allows the movement of water from the lumen of the collecting ducts to the interstitial fluid and then to the bloodstream. This process is essential for maintaining water balance and preventing dehydration [51]. Disruptions in water transport in the renal tubules can lead to conditions like diabetes insipidus, which is characterized by excessive urination and thirst. Diabetes insipidus can be caused by either a deficiency in ADH secretion or resistance to ADH action [52]. In central diabetes insipidus, there is a deficiency in ADH secretion due to a defect in the hypothalamus or pituitary gland. In nephrogenic diabetes insipidus, there is resistance to ADH action due to a defect in the AQP2 channel or downstream signaling pathways [53].

In the lungs, water transport occurs across the alveolar–capillary membrane, which is responsible for gas exchange between the lungs and bloodstream [54]. The movement of water in the lungs is regulated via the oncotic pressure gradient, which is the difference in the concentration of proteins between the plasma and interstitial fluid. The plasma proteins create an osmotic gradient that pulls water from the interstitial fluid to the plasma [47]. This process is essential for maintaining the normal volume of fluid in the lungs and preventing the accumulation of fluid. Disruptions in the oncotic pressure gradient can lead to the accumulation of fluid in the lungs, leading to pulmonary edema [55]. Pulmonary edema is a condition that causes an abnormal buildup of fluid in the air sacs (alveoli) of the lungs, making it difficult to breathe [33]. Pulmonary edema can be caused by various factors, including heart failure, lung injury, or exposure to high altitudes [56]. In heart failure, there is an increase in the pressure in the blood vessels of the lungs, leading to the accumulation of fluid in the interstitial space and alveoli.

In the gastrointestinal tract, water transport occurs in the small and large intestines. The small intestine is responsible for the absorption of water, while the large intestine is responsible for reabsorbing the remaining water and concentrating the fecal matter [57]. The movement of water in the intestines is regulated via the osmotic gradient created by the solutes present in the luminal contents. The absorption or reabsorption of water is regulated via various transporters and channels present on the cell membranes (Goodman, 2002) [58]. Disruptions in the absorption or reabsorption of water can lead to conditions like diarrhea or constipation [59]. In summary, water transport is essential for maintaining the body’s fluid balance and normal physiological functions. Disruptions in water transport can lead to various conditions, including dehydration, pulmonary edema, diarrhea, or constipation. Understanding the pathophysiology of water transport in different tissues is essential for the diagnosis and treatment of these conditions.

## 7. Nutrient Transport

The human body requires nutrients to maintain proper physiological functions, and nutrient transport is crucial for the absorption of these essential molecules. Nutrient transport in the intestine and liver involves a complex interplay of various transporters and channels that facilitate the absorption and processing of different nutrients [60]. In the intestine, the absorption of nutrients such as carbohydrates, proteins, and lipids occurs through the enterocytes of the small intestine (Figure 2). These enterocytes are equipped with specialized transporters that facilitate the uptake of different nutrients. For instance, glucose and galactose are transported into the enterocytes via sodium–glucose cotransporters (SGLTs), whereas fructose is transported through facilitated diffusion via GLUT5 transporters [61]. Proteins are transported via amino acid transporters, and lipids are transported via fatty acid transporters, such as CD36 and FABPpm [62]. After absorption, nutrients are transported to the liver via the portal vein.

In the liver, the processing and storage of nutrients occur through various metabolic pathways. For instance, glucose is stored as glycogen, and excess glucose is converted to triglycerides and transported to adipose tissues for storage [63]. Proteins are metabolized, and their amino acids are used for gluconeogenesis, urea synthesis, and protein synthesis [64]. The liver also plays a crucial role in lipid metabolism, including the synthesis and secretion of lipoproteins and the metabolism of cholesterol [65]. Disruptions in nutrient transport can lead to malabsorption syndromes and liver diseases. Malabsorption syndromes are a group of disorders characterized by impaired nutrient absorption, leading to nutritional deficiencies. For instance, celiac disease is a malabsorption syndrome caused by an autoimmune response to gluten, causing damage to the intestinal villi and impairing nutrient absorption [56]. Similarly, cystic fibrosis is a genetic disorder that affects the transport of ions and nutrients across the intestinal epithelium, leading to malabsorption [66].

**Figure 2 cells-12-02455-f002:**
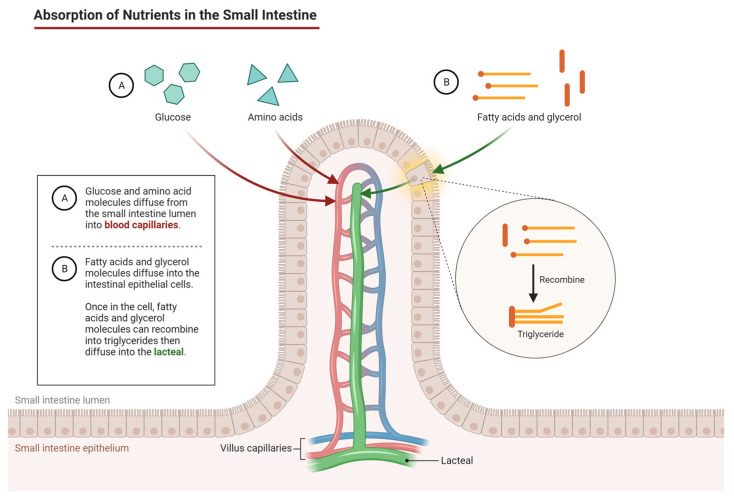
Absorption of nutrients in the small intestine (created by Lim (2020)) [67].

Liver diseases can result from disruptions in nutrient transport, which can lead to malnutrition, inflammation, and other health issues. The liver plays a crucial role in the metabolism and storage of nutrients, including carbohydrates, lipids, and proteins, and any disruption in this process can have a significant impact on overall health. For instance, non-alcoholic fatty liver disease (NAFLD) is a condition characterized by the accumulation of triglycerides in the liver, leading to liver damage [68]. NAFLD is associated with insulin resistance, which impairs the transport of glucose and lipids in the liver [69]. Similarly, Wilson’s disease is a genetic disorder that affects the transport of copper in the liver, leading to liver damage [70]. According to a study by Jou et al. (2018), the disruption of nutrient transport caused by a high-fat diet can contribute to the development of NAFLD. The authors suggest that increased fatty acid uptake by the liver and decreased lipid export contribute to the accumulation of fat in the liver, leading to NAFLD. The dysfunction of nutrient transporters in the liver can lead to the accumulation of toxic metabolites, leading to liver cancer. The authors suggest that restoring the proper function of nutrient transporters may be a potential therapeutic strategy for preventing or treating liver cancer [71].

Nutrient transport is essential for the body’s growth, development, and maintenance. The process involves the digestion of food, absorption of nutrients, and distribution of the nutrients throughout the body. Disruptions in nutrient transport can lead to malabsorption syndromes, which are associated with nutritional deficiencies and other health problems. For instance, celiac disease is a malabsorption syndrome that results from the body’s inability to digest gluten, which is found in wheat, barley, and rye. The condition causes damage to the small intestine, making it difficult for the body to absorb nutrients properly [72]. Another malabsorption syndrome is lactose intolerance, which results from the body’s inability to digest lactose, a sugar found in milk and dairy products. The condition is characterized by abdominal pain, bloating, and diarrhea, and it can be managed by avoiding lactose-containing foods or taking lactase supplements [73]. The absorption of nutrients is a complex process that involves various factors, including the presence of digestive enzymes, the integrity of the intestinal lining, and the functioning of the immune system. Disruptions in any of these factors can lead to malabsorption syndromes. For instance, some medications can interfere with the absorption of nutrients, such as antibiotics that disrupt the gut microbiota [74]. Proper nutrient transport is vital for the body’s proper functioning and overall health. A balanced diet that provides all of the essential nutrients can help to optimize nutrient absorption. Additionally, probiotics and digestive enzymes can aid the absorption of nutrients, particularly for individuals with gastrointestinal disorders.

Nutrient transport is crucial for maintaining proper physiological functions, and disruptions in this process can lead to malabsorption syndromes and liver diseases. Understanding the pathophysiology of nutrient transport in different tissues is essential for the diagnosis and treatment of these conditions.

## 8. Hormonal Regulation of Epithelial Transport

Epithelial transport is essential for maintaining proper fluid and electrolyte balance in the body. Hormones play a critical role in regulating epithelial transport by influencing the transport proteins and channels present in the epithelial cells. The hormonal regulation of epithelial transport is a complex process that involves multiple hormones, including antidiuretic hormone (ADH), aldosterone, and parathyroid hormone (PTH). ADH, also known as vasopressin, is produced in the hypothalamus and released from the posterior pituitary gland [75]. It regulates water balance by increasing water reabsorption in the kidneys and reducing urine output. Aldosterone, produced in the adrenal glands, regulates sodium and potassium balance by increasing sodium reabsorption and potassium secretion in the kidneys [76]. PTH, produced in the parathyroid glands, regulates calcium and phosphate balance by increasing calcium reabsorption in the kidneys and stimulating bone resorption [77].

The hormonal regulation of epithelial transport also involves interactions between hormones. For example, aldosterone and ADH can have additive effects on water reabsorption in the kidneys [76]. PTH can also interact with vitamin D, another hormone, to regulate calcium and phosphate balance [78]. Abnormalities in hormone production or signaling can lead to disorders of epithelial transport. For example, diabetes insipidus is a condition in which the body cannot properly regulate water balance due to a lack of ADH production or signaling [71]. This leads to excessive urine output and dehydration. Hyperaldosteronism, a condition in which the body produces too much aldosterone, can lead to excessive sodium reabsorption and potassium secretion, resulting in high blood pressure and electrolyte imbalances [79].

In addition to ADH, aldosterone, and PTH, other hormones play a role in regulating epithelial transport. For example, the hormone atrial natriuretic peptide (ANP), which is produced by the heart, acts on the kidneys to increase sodium and water excretion, thereby reducing blood volume and blood pressure [80]. Another hormone, namely calcitonin, produced by the thyroid gland, regulates calcium balance by inhibiting bone resorption and increasing calcium excretion in the kidneys [81]. The hormonal regulation of epithelial transport is also influenced by external factors, such as dietary intake and physical activity. For example, low sodium intake can stimulate aldosterone production, leading to increased sodium reabsorption in the kidneys [82]. Physical activity can also affect hormone levels and electrolyte balance. Exercise-induced sweating, for example, can lead to electrolyte losses that need to be replenished through dietary intake [76].

Disorders of epithelial transport can also result from abnormalities in hormone signaling pathways. Abnormalities in epithelial transport can result in a variety of disorders, including cystic fibrosis, renal tubular acidosis, and congenital chloride diarrhea. While genetic mutations and environmental factors have been known to contribute to these disorders, recent research has highlighted the role of hormonal signaling pathways in regulating epithelial transport [83]. For example, primary hyperparathyroidism is a condition in which the parathyroid glands produce too much PTH, leading to excessive bone resorption and calcium reabsorption in the kidneys. In contrast, hypoparathyroidism, a condition in which the parathyroid glands produce too little PTH, can lead to hypocalcemia and neuromuscular symptoms [84].

Hormones, such as aldosterone, vasopressin, and the parathyroid hormone, play a crucial role in regulating the transport of ions and water across epithelial membranes [85]. Abnormalities in the signaling pathways of these hormones can result in a variety of disorders affecting epithelial transport. For example, mutations in the aldosterone receptor gene can lead to the kidneys being unable to reabsorb sodium and chloride ions, resulting in renal tubular acidosis type 2 [85]. Similarly, mutations in the vasopressin receptor gene can result in the kidneys being unable to conserve water, leading to diabetes insipidus [86].

In addition to genetic mutations, environmental factors can disrupt hormone signaling pathways, leading to disorders of epithelial transport. For example, exposure to heavy metals, such as lead and cadmium, has been shown to disrupt the signaling pathways of aldosterone and vasopressin, leading to impaired ion and water transport across epithelial membranes [87]. Recent advances in our understanding of the role of hormone signaling pathways in regulating epithelial transport have led to the development of new therapies for these disorders. For example, drugs that target the aldosterone receptor have been developed for the treatment of hypertension and heart failure [30]. Additionally, vasopressin receptor antagonists have been developed for the treatment of hyponatremia and heart failure [86].

Hormones play a critical role in regulating epithelial transport and maintaining fluid and electrolyte balance in the body. The hormonal regulation of epithelial transport is a complex process that involves multiple hormones and their interactions. Abnormalities in hormone production or signaling can lead to disorders of epithelial transport, highlighting the importance of proper hormone regulation for overall health and well-being.

## 9. Genetic Disorders of Epithelial Transport

Several genetic disorders have been highlighted due to their roles in epithelial transport, compromising the carriage of ions, water, and different kind of substances through cell membranes, leading to different pathologies, such as cystic fibrosis, Bartter syndrome, and Gitelman syndrome.

In this line, cystic fibrosis (CF) has been highlighted by the previous literature as a well-known disease characterized by an important epithelial transport deficiency. Then, CF is a common life-shortening and multisystem condition which involves different organic signs. Firstly, it comprises pulmonary evidence, including chronic and progressive obstructive lung disease, which may compromise gas exchange, generating respiratory distress. It would be explained due to the pathophysiology of CF, as CF involves mucous blockade. Consequently, an increased risk of pulmonary and sinus infections may occur, leading to inflammation processes. Then, all of these events may lead to airway impediment, enhancing bronchiectasis development, one of the most characteristic signs of CF. Regarding other metabolic systems, CF has been related to obstructive biliary tract disease, triggering liver failure, as well as insulin insufficiency in pancreas, thus eliciting insulin-dependent diabetes mellitus development [88]. CF is an autosomal recessive disorder [89], in which patients showed genetic variants of cystic fibrosis transmembrane conductance regulator (CFTR) gen, which translates the CFTR protein. Thus, the recent literature proposed a classification, in which six different classes of mutations were included, attending to its effect in CFTR. Then, it was determined how different gene variations may compromise synthesis, reduce response, cause folding defects, and produce a reduced level of normal and functional or generate instability in the CFTR protein, as well as impair gating of the CFTR channel or decrease its conductance [90]. 

Previous authors stated that patients must present 2 deleterious CFTR variants to develop CF disease, as it is a recessive disorder [88]. For instance, it has been found that more than 2000 different CFTR variants may exist, with many of them being confirmed to cause or be linked to CF [84], as the specific CFTR mutations which a patient may show could be related to the functioning CFTR protein levels present, being it an important key factor in phenotypic severity and tissue involvement [91]. CFTR protein plays an important role in ion transport since it has a double functioning. Primarily, it has been described how CFTR influences different kinds of ion channels, as it inhibits epithelial sodium channels, triggering a reduction in sodium absorption [92]. Thus, in CF, it has been highlighted that CFTR loses its inhibitory ability, which may enhance sodium levels due to its increased absorption in the lumen. Additionally, it has been pointed out that CFTR reduces permeability to chloride at the luminal surface, as well as modulates bicarbonate transport. This fact may be explained due to its capability-modulating bicarbonate/chloride exchangers’ activity [92]. Furthermore, previous authors proposed that other types of channels may similarly be affected, as CFTR also modulates other chloride-associated channels, including calcium-activated chloride channels and outward-rectifying chloride channels [92]. Then, as CF patients present abnormal CFTR functioning, it could be determined how damaged CFTR may enhance epithelial sodium channels’ activity, triggering increased absorption of sodium. Consequently, this elevated sodium reabsorption promotes an increase in chloride transport through other non-CFTR channels, resulting in a net rise in sodium chloride absorption and, subsequently, via osmotic gradient, increased water absorption. The general outcome is a volume diminution of the airway surface liquid, as well as a reduction in airway lumen pH. As a result, a dense and adherent viscous mucus, as well as a reduction in the activity of endogenous pH-dependent antimicrobial peptides, may lead to diminished mucociliary clearance and promote inflammatory and infectious events and bronchiectasis [93,94]. 

Regarding Bartter syndrome and Guitelman syndrome (BS and GS, respectively), both could be considered as different kind of tubulopathies, in which similarities could be found regarding their clinical presentation [95]. In this line, BS and GS would be related to a congenital salt-losing tubulopathies, promoting a disturbance in the thick ascending limb of Henle loop in case of BS, as well as in the distal convoluted tubule as in case of GS. As a result, renal salt wasting added to the increase in the excretion of high levels of potassium through urine, leading to hypokalemic and hypochloremic metabolic events, may trigger alkalosis. Additionally, they could be associated with secondary hyperaldosteronism and low blood pressure [96]. In general conditions, in the thick ascending limb of the Henle loop, an important cotransporter is constantly working. Thus, the furosemide-sensitive sodium-potassium-2-chloride cotransporter (NKCC2) plays a key role in active sodium and chloride uptake by tubular cells. Once sodium is located inside the cells, it is actively pumped outside the thick ascending limb cells by the basolateral Na-K-ATPase, while chloride is released from the cell basolateral through specific chloride channels (ClC-Ka and ClC-Kb). Moreover, in common conditions, the distal convoluted tubule is where sodium chloride reabsorption occurs through another important cotransporter. Thus, the thiazide-sensitive sodium–chloride cotransporter (NCCT) works to improve sodium chloride transport, in combination with the activity of the basolateral Na-K-ATPase. In this case, a sodium intracellular uptake occurs in combination with chloride absorption, and later, chloride is basolaterally released from the cell through ClC-Kb [93]. In healthy patients, NKCC2 works properly, and combined with the sodium/potassium pump, both have the ability to reabsorb approximately 30% of the glomerular-filtered sodium [94]. Bartter syndrome involves several genetic defects, which may strongly modulate sodium transport in the thick ascending limb [96]. The recent literature suggests that its classification could be established by considering the five known causal gene defects [97]. Type 3, classic Bartter syndrome, involves a mutation in CLCNKB, the gene associated with chloride channel-Kb (ClC-Kb) [98]. Failings in the ClC-Kb channel constitute net sodium chloride reabsorption in the thick ascending limb of Henle loop, as well as a rise in sodium chloride transport to the distal tubule, triggering salt wasting, dehydration, and an increase in renin–angiotensin–aldosterone system activity, all of which is associated with hypokalemic metabolic alkalosis [98]. Regarding Gitelman syndrome, it has been shown that it is caused by deactivating mutations in the SLC12A3 gene, which is responsible for thiazide-sensitive sodium–chloride cotransporter (NCCT) activity in the distal convoluted tubule [99]. These mutations lead to a reduction in sodium chloride reabsorption, triggering mild hypovolemia and stimulation of renin–angiotensin–aldosterone system activity. Furthermore, GS is associated with hyperaldosteronism, which may increase sodium reabsorption in the cortical collecting duct through the epithelial sodium channel and, subsequently, increase the secretion of potassium and hydrogen ions, producing hypokalemic metabolic alkalosis [100].

## 10. Pathophysiology of Diarrhea

Diarrhea could be defined as “the passage of 3 or more loose or liquid stools per day, or more frequently than is normal for the individual”, according to the World Health Organization (WHO) [101]. Diarrhea could also be categorized into different types, depending on duration. Thus, if it lasts less than 14 days, it could be classified as acute diarrhea. In contrast, if its duration lasts between 14 and 28 days, it could be considered as persistent diarrhea, whereas if it lasts more than 28 days, it could be defined as chronic diarrhea [102].

Regarding its etiology, diarrhea could be categorized into four types, namely watery, fatty, inflammatory, and functional [103]. In turn, these types of diarrheas can be further subdivided according to other classifications. Then, watery diarrhea could be further divided into either osmotic or secretory types. In this line, secretory diarrhea is triggered via decreased water absorption and leads to raised stool volumes. In contrast, osmotic diarrhea is produced via water retention in the intestine due to the presence of osmotic active substances that have been poorly absorbed in gut [44]. Regarding fatty diarrhea, it could be defined as a diarrhea in which malabsorption disorder could be present, and subsequently, it presents feces characterized by a high content of fatty substances, known as steatorrheic feces [103]. Related to inflammatory diarrhea, it is characterized by the presence in the feces of several substances related to inflammation and damage, such as leukocytes and blood, as well as different proteins, such as lactoferrin or calprotectin [44]. Usually, it is described as ulcerative colitis and Crohn disease, otherwise known as inflammatory bowel diseases, and they are characterized by a dysregulated mucosal immune response, promoting inflammation processes associated with weight loss, abdominal pain, and, in many cases, bloody diarrhea [104]. In this line, irritable bowel syndrome is characterized by abdominal pain, generally associated with defecation, and modifications to stool frequency, as well as its form or appearance, occurring at least once per day or week over 3 months [105]. Finally, functional diarrhea could be defined as chronic diarrhea, in which the number of evacuations is four or more and they are large and unformed, if patients do not present pain and its onset occurs in the early stages of life, such as childhood [106]. Moreover, the absence of abdominal pain or swelling is what differentiates functional diarrhea from irritable bowel syndrome [44]. 

Concerning its physiopathology, diarrhea also could be defined based on its pathological pathways, since different types of diarrhea could be produced by irregularities that involve different pathways. In this line, abnormalities in epithelial transport may have an important role in pathophysiology. Thus, in general conditions, the combined absorption of sodium and chloride ions is a fundamental mechanism that is mediated through two types of transporters located in the apical membrane of epithelial cells, namely solute carriers (SLC). Thus, SLC9 is responsible for sodium–hydrogen exchange, whereas SLC26 is liable for chloride–bicarbonate exchange. Additionally, sodium and chloride could be transported to the basolateral membrane through the Na+/K+ ATPase and a potassium chloride co-transporter, respectively [14,107]. Additionally, it has been found that sodium could be absorbed without concomitant chloride recapture in the distal colon. Thus, sodium may enter the apical membranes of cells through the epithelial sodium channel, and consequently, it could be exported across the basolateral membrane into Na+/K+ ATPase [108]. Regarding chloride, its secretory mechanism comprises an uptake across the basolateral membrane through the sodium–potassium-2 chloride co-transporter (NKCC1), partially mediated by Na+/K+ ATPase. Then, when intracellular sodium levels rise, they trigger an extrusion of sodium, which promotes NKCC1 activity. Additionally, this process is mediated though potassium ions, which may also be reused from the basolateral membrane by different kind of channels, including cyclic AMP and calcium-activated channels, with the aim being to preserve the positive electrical gradient that allows chloride to leave across the apical membrane [109]. Hence, the activity of intestinal SLC9, SLC26, epithelial sodium channels, and NKCC1 transporters could be downregulated via a variety of pro-inflammatory cytokines, which may compromise diarrhea volume both in acute diarrhea, due to infections by microorganism or by toxins activity, and patients who suffer from inflammatory bowel diseases [109,110,111,112]. For example, it has been shown that rotavirus introduces a non-structural glycoprotein, namely NSP4, into the intestinal lumen, which may have a negative impact on the epithelial sodium channel by inhibiting its activity [110]. Furthermore, rotavirus infection also enhances active chloride secretion through calcium-activated chloride channels, promoting ion dysregulation, thus triggering acute diarrhea [111]. Finally, mice models proposed how different infections may stimulate diarrhea processes, since colitis produced by enteropathogenic *E. coli* and *C. rodentium* were both associated with decreased expression of SLC26, suggesting the possible causative effects that may cause toxins and pro-inflammatory cytokines to be released from or in presence of these microorganisms in these cotransporters, affecting ion exchanges [112,113]. Finally, regarding chronic diarrhea, it has been proposed that the different mutations that affect nucleotide polymorphisms may compromise SLC9A3 expression, which may raise susceptibility to chronic diarrhea, since it could decrease the activity of SLC9 transporter [114]. Regarding secretory diarrhea, it has been proposed that they could be affected by a dysregulation in CFTR, mentioned in cystic fibrosis section. CFTR is present in the apical membrane of enterocytes, and it is known for its activity in chloride secretion, especially in cyclic AMP-mediated diarrhea, as in the case of cholera, promoting the liberation of liquid into the intestinal lumen [115], enhancing diarrhea events. Concerning inflammatory diarrheas, the previous literature suggested that inflammatory substances may compromise the expression and activity of numerous SLC26 isoforms [116]. Furthermore, it has been shown by the recent literature that both Japanese and Chinese cohorts presented polymorphisms in the SLC26A3, recognized as a risk factor for ulcerative colitis [117]. In this case, the importance of SLC26A3 in colonic electrolyte and fluid balance is crucial, as it is downregulated, improving chloride secretion and, subsequently, causing watery diarrhea [118].

According to these findings, the importance of these cotransporters in ion exchanges has been proven, promoting solute dysregulation and, subsequently, enhancing diarrhea events.

## 11. Pathophysiology of Hypertension

The world’s highest cardiovascular morbidity and mortality rates continue to be associated with hypertension. Although numerous factors contribute to poor arterial blood pressure (ABP) control and pseudo resistance (such as ignorance, lifestyle habits, non-adherence to medication, inadequate treatment, drug-induced hypertension, and undiagnosed secondary causes), 10.1% of patients treated for elevated ABP have true resistant hypertension (RH) [119,120]. Despite the use of multiple treatment modalities, blood pressure control to guideline-recommended target levels is often not achieved [121].

It is highly possible that many inter-related factors contribute to hypertensive patients’ elevated BP, and their relative contributions may vary between individuals. Among the factors that have been thoroughly examined include salt intake, obesity, insulin resistance, the renin–angiotensin system, and the sympathetic nervous system. The following examples are the physiological processes typically involved in the emergence of essential hypertension: cardiac output, peripheral resistance, autonomic nervous system, renin–angiotensin system, and endothelial dysfunction (Figure 3). Other factors, such as genetics, low birth weight and intrauterine nutrition, and neurovascular anomalies, have been also evaluated in recent years [122].

Harmony between cardiac output and peripheral vascular resistance is necessary to maintain normal blood pressure. The small arterioles, which have smooth muscle cells on their walls, are what determine peripheral resistance, rather than the large arteries or capillaries [123]. Long-term smooth muscle constriction is thought to cause structural changes, with the thickening of the artery walls causing an increase in peripheral resistance that cannot be reversed [124]. In very early hypertension, it has been argued that peripheral resistance does not develop and the rise in blood pressure is due to an increase in cardiac output, which is associated to sympathetic overactivity [125]. The ensuing rise in peripheral arteriolar resistance may develop as a compensatory mechanism to prevent the elevated pressure from being conveyed to the capillary bed, where it would have a substantial impact on cell homeostasis. According to Fagard and Staessen [126], hypertensive people under the age of 25 had an increased cardiac output, whereas older subjects had a normal cardiac output. It is now known that long-term pathophysiological manifestations of hypertension that raise vascular tone and resistance exist in addition to systemic autoregulation, which is a physiological response [119].

As mentioned, arteriolar constriction and dilatation can both be caused by sympathetic nervous system stimulation. As a result, the autonomic nervous system plays a critical role in maintaining normal blood pressure. Although recent research on the sympathetic nervous system’s role in hypertension, such as da Silva et al.’s study, have concentrated on brain pathways [127] or associated signaling mechanisms that increase or decrease ABP [128], parallel studies have been conducted to determine which sympathetic nerves become hyperactive or whose modulation lowers ABP. Evidence reported that elevated plasma or cerebrospinal fluid NaCl concentrations are one signal thought, initiating and sustaining sympathoexcitation in salt-sensitive hypertension (Figure 3) [129]. Although it is known that the etiology of hypertension is multifactorial, the environmental causes, such as excessive consumption of dietary sodium, has been identified as a well-established contributor. The human population has a range of sodium sensitivity, as certain individuals manifest a substantial increase in blood pressure upon heightened salt consumption, while others undergo minor alterations. The observed variations can be partially attributed to the genetic diversity of the mechanisms responsible for sodium regulation and elimination. With regard to this issue, the epithelial sodium channel (ENaC) plays a crucial role in facilitating the reabsorption of sodium inside the distal nephron [130]. The channel in question plays a significant role in the control of the amount of extracellular fluid, hence impacting blood pressure levels. The observed diversity in sensitivity to sodium ions can be attributed, at least in part, to genetic variations associated with the renin–angiotensin–aldosterone system and renal sodium transporters. Salt-sensing variability is influenced by several factors, including immune cells, vascular endothelium, smooth muscle, and particular parts of the central nervous system (CNS) [131].

In this regard, rodents’ studies also reported outcomes that revealed that a lesion of the anteroventral third ventricular region of the hypothalamus may attenuate the development or severity of multiple experimental models of salt-sensitive hypertension [132]. Kopp et al. reported that high salt intake or aging-related tissue sodium accumulation has been linked to hypertension and an increased risk of cardiovascular disease [133]. Recently, it has been proposed that the immunomodulatory effects of salt deposited in the skin may have evolved to serve as a barrier against microbial invasion [134]. However, other recent studies indicate that a very high sodium intake can trigger prototypical metabolic changes aimed at water conservation, with the side effect of increasing blood pressure [135].

Evidence also reports that a blockade of sympathetic outflow or sympathetic nerve transection lowers ABP [136] and the interruption of neurotransmission in several sympathetic-regulatory nuclei lowers, and it even normalizes sympathetic nerve activity (SNA) and/or ABP [137]. Increased blood pressure variability predisposes people to end-organ damage and the onset of cardiovascular disease, so this observation has important clinical implications. Increased blood pressure variability is a reliable indicator of later cardiovascular disease and adverse events. It has not been established how a high-salt diet “sensitizes” central autonomic networks without altering baseline SNA or ABP. However, lowering salt intake or administering diuretics like thiazides are successful treatments for high blood pressure. In human populations, sodium sensitivity is a relatively common phenotype linked to an increased risk of hypertension and cardiovascular events. It is defined as an exaggerated change in blood pressure in response to extremes of dietary salt intake [138,139,140].

Concerning the renin–angiotensin–aldosterone system (RAAS), it may be the most important endocrine system regulating blood volume and vascular resistance affecting blood pressure regulation and cardiac output [141]. The juxtaglomerular apparatus of the kidney secretes renin in response to the hypoperfusion of the glomeruli or a reduction in salt intake. It is also secreted in response to the activation of the sympathetic nervous system [142]. Renin serves as the catalyst for converting renin substrate (angiotensinogen) into physiologically inactive angiotensin I, which is promptly converted to physiologically active angiotensin II by an angiotensin-converting enzyme in the lungs (ACE). Angiotensin II, a powerful vasoconstrictor, induces a rise in blood pressure. In addition, it promotes the secretion of aldosterone from the zona glomerulosa of the adrenal gland, resulting in a surge in blood pressure associated with sodium and water retention (Figure 2) [143]. Although some studies reported that the circulating renin–angiotensin system is not believed to be directly responsible for the rise in blood pressure [122], one of the main causes of several types of hypertension has been identified as the disruption of the renin–angiotensin system (Figure 3) [144]. Many hypertension patients have traditionally had low levels of renin and angiotensin II, and medicines that block the renin–angiotensin system have not been particularly successful [122]. However, it is demonstrated that the RAAS plays a significant role in controlling the inflammatory response [145]. This system encourages the production of several cytokines, including IL-6, TNF-a, and COX-2, and aids in the recruitment of inflammatory cells to the site of injury. Additionally, there is a link between RAAS and oxidative stress, another significant factor involved in atherogenesis [146].

In relation to the preceding point, by producing several potent local vasoactive substances, such as the vasodilator molecule nitric oxide and the vasoconstrictor peptide endothelin, vascular endothelial cells play a crucial role in cardiovascular regulation [147]. In this regard, Gimbrone et al. specified that in its broadest sense, endothelial cell dysfunction refers to a collection of different non-adaptive changes in functional phenotype that have significant effects on the control of thrombosis and hemostasis, local vascular tone and redox balance, and the orchestration of acute and chronic inflammatory responses within the arterial wall [148]. Consequently, human essential hypertension has been linked to endothelium dysfunction. Nitric oxide (NO), a vasodilator, and angiotensin II, a vasoconstrictor, are released by the vascular endothelium to maintain tone under physiological conditions [149]. Reactive oxygen species (ROS) of pro-inflammatory cytokines, such as interleukin 1 (IL1) and tumor necrosis factor (TNF)-a, are abnormally produced, and NO is released less frequently, resulting in endothelial dysfunction (Figure 3). Additionally, Wu et al. also reported that ROS can increase vasoconstriction, which will change vascular caliber and reactivity. Hence, ROS are crucial for the remodeling of micro arterioles and vascular stiffening. This condition initiates the development of atherosclerosis. Other conditions caused by ROS can be seen in the renal system promoting salt reabsorption in the kidney, hence increasing cardiac output and volume. ROS also boost neuronal activity in important brain centers, such as the subcortical organ and brainstem centers, thus affecting neural regulation in the brain [150,151,152].

Multiple genes are likely to play roles in the development of the disorder in a particular person, even though distinct genes and genetic factors have been linked to the onset of essential hypertension. Therefore, it is very challenging to estimate the relative contributions of each of these genes. Highly heritable Mendelian forms of hypertension have been linked to several single-gene mutations, including those causing Liddle syndrome, pseudo hyperaldosteronism, aldosterone-producing adenomas, glucocorticoid remedial hypertension, and missense mutations of the mineralocorticoid receptor. Even though these rare single-gene mutations have shed light on the pathogenesis of hypertension, they cannot account for the high prevalence of familial hypertension that is frequently seen in clinics [153,154].

## 12. Pathophysiology of Edema

Edema is a condition characterized by the accumulation of excess fluid in tissues, either within cells (cellular edema) or in the collagen–mucopolysaccharide matrix of interstitial spaces (interstitial edema) [155]. The present review focuses on interstitial edema, which occurs due to abnormal changes in pressure (hydrostatic and oncotic) across microvascular walls, alterations in molecular structures that form the barrier to fluid, and solute movement in endothelial walls. Examples include changes in the hydraulic conductivity and osmotic reflection coefficient for plasma proteins and disruptions in the lymphatic outflow system, as suggested using the Starling equation [155].

In this regard, the transportation of ions plays a crucial role in the regulation of transmembrane and transcellular electrical potential, fluid transportation, and cellular volume [156]. The disruption of ion transport has been linked to cellular dysfunction, as well as the occurrence of intra- and extracellular edema and irregularities in the amount of epithelial surface liquid [157]. There is a growing body of research suggesting that diseases characterized by a heightened local or systemic inflammatory response are correlated with aberrant ion transport. The pathogenesis of various conditions, such as hypovolemia caused by fluid losses, hyponatremia and hypokalemia in diarrheal diseases, electrolyte abnormalities in early infancy pyelonephritis, septicemia-induced pulmonary edema, and hypersecretion and edema induced via inflammatory reactions in the upper respiratory tract, has been associated with this atypical ion transport [158]. The membranous ion transport systems encompass various components that have been observed to undergo functional alterations in response to inflammation. These components include the sodium potassium ATPase, the epithelial sodium channel, the cystic fibrosis transmembrane conductance regulator, calcium-activated chloride channels, and the sodium potassium chloride co-transporter. Several inflammatory mediators that have an impact on ion transport include tumor necrosis factor, gamma interferon, interleukins, transforming growth factor, leukotrienes, and bradykinin [156].

The detrimental effects of excessive buildup of interstitial fluid on tissue function are widely recognized due to the increased diffusion distance for oxygen and other nutrients. This might possibly compromise cellular metabolism in the swollen tissue [159]. Moreover, the presence of edema restricts the diffusion process, impeding the elimination of potentially harmful byproducts generated during cellular metabolism. The issue at hand presents a significant challenge within the pulmonary system, since the occurrence of pulmonary edema can greatly hinder the process of gas exchange [160]. Certain tissues include structural features that limit the enlargement of interstitial gaps when exposed to edemagenic stress [161]. As an illustration, the kidneys are enveloped inside a resilient fibrous capsule, the brain is encompassed by the cerebral vault, and skeletal muscles within distinct compartments are contained within constrictive fascial sheaths. Consequently, even little increases in transcapillary fluid filtration can lead to substantial elevations in interstitial fluid pressure within these tissues [162]. Consequently, this process results in a decrease in the pressure gradient across capillaries and causes their physical compression, ultimately leading to a reduction in tissue perfusion [163]. Within the intestinal tract, unregulated transcapillary filtration has the potential to lead to the exudation of interstitial fluid into the lumen of the gut. This occurrence is commonly referred to as filtration–secretion or secretory filtration [163]. The absorptive function of the sensitive intestinal mucosa may be compromised via filtration–secretion, which is believed to be caused by the development of wide channels between mucosal cells at the villous tips when the interstitial fluid pressure exceeds 5 mmHg [159]. Ascites is a prevalent complication of cirrhosis characterized by the abnormal buildup of fluid within the peritoneal cavity. This condition arises from the leakage of fluid originating from clogged hepatic sinusoids, a consequence of heightened portal venous pressure. The presence of ascites in individuals can elevate susceptibility to peritoneal infections, hepatic hydrothorax, and abdominal wall hernias [164].

Yet, edema can occur in different forms, including hydrostatic edema, permeability edema, and lymphedema. In this line, hydrostatic edema happens when there is an accumulation of excess interstitial fluid due to elevated capillary hydrostatic pressure [159]. Permeability edema, on the other hand, is caused by disruptions in the physical structure of the pores in the microvascular membrane, making the barrier less effective in terms of restricting the movement of macromolecules from the blood to the interstitial fluid [165]. Lymphedema is a form of edema that can result from various factors, such as impaired lymph pump activity, increased lymphatic permeability, lymphatic obstruction (e.g., microfilarisis), or the surgical removal of lymph nodes, which is often performed in breast cancer treatment. Furthermore, destruction of extracellular matrix proteins, which can occur during inflammation due to the formation of reactive oxygen and nitrogen species and release of hydrolytic enzymes from various cells, including leukocytes, immune cells, and tissue cells, can alter the compliance characteristics of the interstitial gel matrix [166]. This alteration may result in interstitial fluid pressure failing to increase and oppose fluid movement while also diminishing the tensional forces exerted by extracellular matrix proteins on the anchoring filaments attached to lymphatic endothelial cells that facilitate lymphatic filling due to disrupted mechanical integrity. Additionally, reductions in circulating plasma proteins, particularly albumin, can lead to edema by decreasing plasma colloid osmotic pressure. This reduction in plasma proteins commonly occurs in liver disease and severe malnutrition, contributing to the development of edema in these conditions [159].

In addition to the compensatory mechanisms mentioned earlier, there are other safety factors that help to protect against edema in response to increased arterial or venous pressure in certain tissues. One such factor is the myogenic response, which involves arteriolar vasoconstriction in response to increased wall tension, attenuating the rise in capillary pressure that could occur due to arterial or venous hypertension [167]. This response also reduces the available microvascular surface area for fluid exchange by closing precapillary sphincters. Moreover, when venous pressure is elevated, the volume of blood within postcapillary venules, larger venules, and veins increases, causing them to bulge into the extravascular compartment and raise tissue pressure. This venous bulging, in turn, stiffens the extracellular matrix by increasing tensional forces on the reticular fibers and fluid in this space. Additionally, changes in the excluded volume, resulting from increased transcapillary fluid filtration, contribute to the margin of safety against the swelling of the extracellular matrix compartment [159].

Thus, it is evident that tissues with restrictive endothelial barrier properties, low interstitial compliance, and a high sensitivity of lymph flows to changes in interstitial fluid pressure are likely to have a greater margin of safety against edema formation. Even in tissues where the endothelial barrier is less restrictive and lymphatic sensitivity is low, the margin of safety can still be significant if the interstitial matrix is stiff [168]. This highlights the complex interplay between various factors that contribute to the regulation of tissue fluid dynamics and the prevention of edema formation.

Regarding hypoproteinemia, marked reductions in circulating protein levels, particularly albumin, can result in edema related to intravascular factors [168]. Hypoproteinemia, which can be caused by various conditions such as compromised glomerular barrier in diseased kidneys, impaired hepatic synthesis of plasma proteins in liver disease, severe malnutrition, or protein-losing enteropathy, as well as the infusion of intravenous fluids without macromolecules, leads to a reduction in the colloid osmotic pressure gradient (πc − πt) during non-steady-state conditions [169]. This reduction opposes the hydrostatic pressure gradient that favors filtration, resulting in a significant transcapillary flux of protein-poor fluid into the interstitial spaces. Like capillary hypertension, this effect is counteracted by elevated tissue hydrostatic pressure, which increases lymph flow; both of these factors serve to limit tissue fluid accumulation. Increased capillary filtration also acts to dilute protein concentration in the extracellular spaces, further amplified by the increased accessible volume in the extracellular matrix gel [159]. The subsequent reduction in interstitial colloid osmotic pressure decreases net filtration pressure, thereby minimizing edema formation. However, unlike the response to vascular hypertension, there is no stimulus for myogenic arteriolar vasoconstriction and venous bulging in hypoproteinemia, reducing the margin of safety for edema formation in response to this edemagenic stress. Consequently, tissues are less able to compensate for reductions in plasma colloid osmotic pressure equivalent to a given increase in capillary hydrostatic pressure, making hypoproteinemia a more vulnerable condition compared to vascular hypertension in terms of edema formation [159].

Permeability edema and inflammation are often seen pathological reactions in many illness conditions, frequently coinciding with trauma and being initiated by endogenous mediators or pharmacological substances like escin [170]. The Starling equation demonstrates that heightened permeability is manifested via a decrease in the osmotic reflection coefficient and/or an increase in hydraulic conductivity. Consequently, this leads to the diminished efficacy of the colloid osmotic pressure gradient in terms of counteracting filtration. Consequently, there is a resultant transfer of filtrate, rich in proteins, into the interstitial spaces, thus augmenting the colloid osmotic pressure and net filtration pressure inside the tissue. Several mediators that enhance microvascular permeability also function as vasodilators, thus decreasing resistance in arterioles and, subsequently, raising the capillary pressure. Consequently, this leads to an increase in the net filtration pressure [171]. Vasodilatation has the additional effect of enlisting capillaries, enlarging the microvascular surface area so that is accessible for the movement of fluid and proteins into the tissues. This process further augments the capillary filtration coefficient [172]. The augmented transcapillary fluid filtration leads to the heightened convective transportation of proteins across the expanded pores in the microvascular barrier [173]. Nevertheless, in the presence of inflammatory circumstances marked by the infiltration of leukocytes into the tissues, the severity of the permeability edema is heightened. Inflammation is a biological reaction triggered by tissue injury and characterized by the release of mediators that induce microvessel permeability and vasodilation, as well as the recruitment of leukocytes to the site of damage. These particular types of white blood cells secrete enzymes and reactive species that break down the components of the extracellular matrix, which, in turn, impairs the ability of the lymphatic system to fill properly and leads to an increase in the compliance of the interstitial space. This phenomenon permits a greater volume of extracellular fluid to be contained inside the matrix, resulting in little elevation in the interstitial fluid pressure. As a result, the efficacy of the edema safety factor is diminished [174]. Moreover, the breakdown of fibroblast linkages with collagen fibers inside the interstitial spaces leads to a decrease in excluded volumes and an increase in the effective interstitial colloid osmotic pressure. The disturbed matrix further enables the transportation of extravasated proteins from blood to lymph. In general, the involvement of permeability edema and inflammation in the pathophysiology of different illness conditions are multifaceted, resulting in changes to the function of microvascular barriers, fluid filtration, and lymphatic transport [175].

In addition, neurogenic inflammation is a multifaceted physiological reaction marked by leucosequestration, the development of edema, and the extravasation of plasma proteins. This response happens subsequent to the activation of sensory neurons [176]. Upon stimulation, sensory fibers produce a variety of pro-inflammatory chemicals, including calcitonin gene-related peptide, substance P, and neurokinin A. These molecules have the potential to not only exert a direct effect on the microvasculature, leading to the induction of inflammation, but also have the ability to activate tissue mast cells. This activation of mast cells can further enhance the inflammatory response by triggering the release of their own mediators. Recent studies have elucidated that neurogenic inflammation extends beyond its conventional role within the nervous system, including a wider range of physiological activities [177]. For instance, research has indicated that neurogenic inflammation may have a role in the modulation of immune responses and the process of wound healing [178]. Moreover, recent findings indicate that neurogenic inflammation may potentially contribute to the pathophysiological mechanisms underlying chronic pain disorders, including fibromyalgia and neuropathic pain [179]. The increasing comprehension of neurogenic inflammation emphasizes its complex characteristics and the necessity for more investigation to comprehensively clarify its processes and possible treatment implications.

Myxedema is a medical disorder that is distinguished by the atypical buildup of mucopolysaccharides and collagen in the interstitial spaces of tissues. This illness arises due to the excessive production of these extracellular matrix components by fibroblasts [179]. The overproduction of interstitial collagen and mucopolysaccharides results in an elevation in the density of the extracellular matrix. This, in turn, leads to the augmentation of the elastic recoil of the interstitial gel matrix and the generation of a significantly negative interstitial fluid pressure. Consequently, there is a reduction in the flow of lymph and an increase in the excluded volume, denoting the area filled by the matrix components. This ultimately results in an elevation in the effective interstitial colloid osmotic pressure. The aforementioned alterations cumulatively provide a suction effect that enhances the pace of fluid filtration and facilitates the formation of edema. Moreover, scholarly investigations have provided insights into the complex etiology of myxedema, indicating that it is not exclusively confined to instances caused by hypothyroidism [180]. In addition to primary causes, myxedema can be influenced by other underlying variables, including autoimmune disorders, genetic abnormalities, and certain pharmaceutical treatments. Moreover, myxedema can present itself in diverse manners, such as through the occurrence of edema and the hypertrophy of the dermis, notably occurring in the facial, manual, and pedal regions, alongside vocal hoarseness, muscular debility, and alterations in the texture of hair and nails [181]. Recognizing and effectively managing the underlying causes of myxedema are crucial in order to mitigate complications and enhance patient outcomes.

Lymphedema, a pathological disease distinguished by the buildup of interstitial fluid and macromolecules as a result of impaired lymphatic channel function, can be attributed to a range of causative events. Lymphedema can occur as a result of various factors that impede the normal functioning of the lymphatic vessels. These factors include the physical blockage of the vessel lumen caused by tumors or the spread of tumor cells, the destruction or regression of pre-existing lymphatics, the ineffectiveness of valves between lymphangion, the paralysis of lymphatic muscles, reduced tissue motion, diminished arterial pulsation or venomotion, and increased venous pressure at the drainage points [182]. It is important to acknowledge that the occurrence of edema development due to lymphatic dysfunction is contingent upon a reduction in lymph flow of 50%, providing that all other parameters stay the same [159]. When there is a full obstruction, the flow of lymph in the afflicted tissue region ceases, resulting in the persistence of transcapillary filtration until the interstitial pressure increases to equal the net filtration pressure. The reduction in transcapillary volume flow results in a concomitant decrease in the transfer of protein from the vascular compartment to the interstitial compartment. Nevertheless, due to the impaired lymphatic system’s inability to eliminate extravasated proteins, the diffusion of protein persists until the concentration gradient is depleted. In a state of equilibrium, the interstitial fluid pressure increases to align with the microvascular hydrostatic pressure. Additionally, the interstitial colloid osmotic pressure is equal to the plasma colloid osmotic pressure, leading to a net filtration pressure of zero [183]. Consequently, there is a notable augmentation in the levels of water and protein, as well as the development of fibrosis and deposition of fat cells inside the impacted tissue. Once again, it has been demonstrated by researchers that genetic abnormalities, infections, trauma, and inflammation can all play roles in the development of lymphatic dysfunction and eventual lymphedema. Furthermore, the progression of imaging methodologies and the use of lymphatic mapping have significantly enhanced our comprehension of the lymphatic system and its significance in the context of lymphedema. Moreover, the ramifications of lymphedema on individuals extend beyond mere physical manifestations, as it may profoundly influence their overall well-being, resulting in sensations of unease and distress and constraints on their daily activities. The timely identification and implementation of effective treatment approaches, including manual lymphatic drainage, compression therapy, and exercise, play pivotal roles in mitigating problems and enhancing outcomes for those afflicted with lymphedema [184,185,186].

## 13. Pathophysiology of Renal Disease

Renal functions include blood filtration, the metabolism and excretion of endogenous and exogenous compounds, and endocrine functions. Most importantly, the kidneys are the primary regulators of fluid, acid–base, and electrolyte balance in the body, and this remarkable pair of organs maintains homeostasis in response to a wide range of dietary and environmental changes [187]. Kidney diseases are divided into two main categories, namely acute kidney injury (AKI) and chronic kidney disease (CKD). CKD is far more prevalent worldwide than previously believed. It affects 10 to 15% of the adult population in Western nations, many of whom require expensive treatments or dialysis [188]. In this regard, one of the primary concerns is its late diagnosis, which is only possible when the disease has progressed. The absence of a clinical manifestation in the early stages and the fact that the commonly measured parameters of renal function are only significantly reduced in the late stages of the disease are the primary causes of this problem [189].

One objective for researchers investigating AKI should be to develop a comprehensive theoretical framework that effectively integrates the molecular and inflammatory ramifications of sepsis with compromised epithelial tubular function, reduced glomerular filtration rate (GFR), and eventual renal failure. Recent research findings have provided insights into the mechanisms via which sepsis affects the control of ion, glucose, urea, water transport, and acid–base homeostasis in the renal tubules [190]. In sepsis, inflammation is a significant physiological mediator that can have a detrimental impact on the complex tubular systems involved in regulating sodium, potassium, and chloride levels, even when cellular perfusion is normal. All of the mentioned deficits have the potential to result in the elevated transport of chloride to the distal tubule, hence compromising GFR and overall renal function. Schmidt et al. [191] conducted a study that demonstrated a significant decrease in the expression of various renal transport proteins, including Na+/H+ exchanger 3 (NHE3), Na+/K+-ATPase, renal outer medullary K+ channel (ROMK), ENaC, Na+–K+–2Cl−-cotransporter 2 (NKCC2), Na+–Cl− cotransporter (NCC), and kidney specific chloride channel -1 and -2 (CLCK-1 and -2), as well as Barttin, in an in vivo model of rat sepsis induced via the intraperitoneal injection of liposaccharide (LPS). The aforementioned discovery has been autonomously replicated using similar models, which have demonstrated the downregulation of NKCC2 and NHE3 [192]. Also, several mediators have been suggested as potential links between systemic inflammatory mediators, like TNF-α, IL-1, and liposaccharide (LPS), and the impairment of the epithelial transport function [193]. However, substantial evidence supports the direct involvement of cyclo-oxygenase and prostaglandins, particularly in the mediation of distal sodium and chloride delivery. In the majority of these tests, the administration of cyclo-oxygenase inhibitors, such as indomethacin, had a mitigating effect on salt wasting in hyperinflammatory models of renal injury [194].

Additionally, mitochondria are the primary source of oxygen radicals, which increase the kidneys’ vulnerability to oxidative stress-induced damage. Both reactive oxygen species (ROS) and reactive nitrogen species (RNS) have been implicated in a variety of signaling pathways that regulate cell growth and differentiation, mitogenic responses, extracellular matrix production and breakdown, apoptosis, oxygen sensing, and inflammation. ROS and RNS contribute to the immune system’s defense against pathogenic microorganisms, in addition to their regulatory function. Nevertheless, their involvement in renal diseases was predictable [195,196]. Produced during acute or chronic kidney injury, free radicals and pro-oxidants may exacerbate the progression of the disease and play a role in the pathogenesis of subsequent complications [197]. Some serious symptoms in distant organs, such as cardiovascular diseases and neurological dysfunction, are frequently observed in AKI and CKD (Figure 4) [198,199].

Significant information regarding the pathophysiology of AKI is presented in recent reviews such as those by Pickkers et al. and Peerapornratana et al. [200,201]. It has been specified that AKI is characterized by a rapid loss of kidney function (within a week) that leads to the accumulation of toxic end products of nitrogen metabolism and creatinine in the blood, decreased urine output, or both [202]. In hospitalized patients, studies such as those conducted by Wu et al. have attempted to establish that identifying symptoms preceding AKI leads to a larger possibility of earlier detection of the disease. For instance, they have identified low blood pressure (below 80) and medications including tazobactam and vancomycin as potential causes [203]. However, despite all efforts, kidney failure occurs for various reasons. Thus, perennially, as a result of decreased renal blood flow; renally, as a result of damage to the renal parenchyma; or postrenally, as a result of obstruction of urine flow from the renal collecting system or ureters, are three ways that AKI can manifest (Figure 4) [204]. This type of prerenal AKI may or may not result in ischemic acute tubular necrosis. However, both prerenal (hemodynamic changes, endothelial dysfunction) and intrarenal (inflammatory infiltration and renal parenchymal damage, intraglomerular thrombosis, and tubular obstruction) mechanisms contribute to sepsis-induced kidney injury, but the exact sequence of events is not fully understood. Nevertheless, it is well established that ROS are involved on multiple levels [202] and on a cellular level, and the pathogenesis of AKI is characterized by intricate interactions between inflammatory immune cells and renal cell mediators [204]. When studying this process in rodents, Zhao et al. demonstrated that reducing ROS levels in AKI animals reduced renal failure, mitochondrial damage, and inflammation [205]. In this regard, Pavlakou et al. pointed out that inflammation-induced upregulation of inducible nitric oxide synthase results in the production of excessive nitric oxide (NO), which, in turn, uncouples endothelial NO synthase, resulting in the production of highly reactive super oxides through oxygen oxidation [204]. Moreover, ischemia and reperfusion are also activators of oxidative stress, with mitochondria being the primary source of ROS in this setting [205]. ROS such as the hydroxyl radical, peroxynitrite, and hyperchlorous acid are also produced during ischemia damage. Antioxidant enzymes such as superoxide dismutase (SOD), catalase, and glutathione reductase are also deficient. This has been demonstrated in renal tissue following ischemia and nephrotoxicity [206,207]. Zarbock et al. have previously mentioned that excess NO interferes with SOD and combines with superoxide radicals to produce peroxynitrite, which has a direct detrimental effect on tubular epithelial cells. (TECs) [208]. Thus, evidence revealed that inflammatory cells are notable for employing ROS as part of their mode of operation [209]. Dendritic cells and neutrophils are the primary contributors to renal damage in septic AKI. However, recent research demonstrates that inhibiting NOX2 and inducible NO synthase in neutrophils reduces kidney damage in a mouse model of sepsis-induced AKI [210] (Table 1). The knowledge of the pathophysiology and the quality of diagnostic techniques for simultaneous kidney and liver impairment are currently inadequate. It is becoming increasingly apparent that inflammation is a significant cause of AKI, particularly in individuals with infection and multiorgan failure [211]. As AKI in chronic liver illness is persistent and becomes increasingly irreversible with medical treatment as the condition progresses, newer medicines and new techniques for either liver or renal support are necessary.

In a similar vein, chronic kidney disease (CKD) is commonly attributed to underlying conditions, such as diabetes, hypertension, glomerulonephritis, or polycystic kidney disease. The presence of these many biological processes in CKD patients may be facilitated by the heightened levels of oxidative stress, as seen in Figure 4 [212]. As the progression of deterioration advances, the fibrosis process not only affects the kidney but also impacts the heart, resulting in significant cardiac dysfunction, irrespective of the underlying cause. One of the most robust indications of progression to chronic kidney disease (CKD) is renal fibrosis, as seen in Figure 4 [213]. Various discrepancies in illness progression and mortality based on gender have been observed. Specifically, males exhibit higher rates of mortality from all causes and cardiovascular diseases, as well as an elevated risk of developing chronic kidney disease (CKD), compared to women. CKD is a significant worldwide health issue that is becoming more prevalent [214]. The interplay between oxidative stress, chronic inflammation, and endothelial dysfunction is recognized as a key factor in the ongoing reciprocal relationship between chronic kidney disease (CKD) and systemic diseases [215]. Therefore, the progression of chronic kidney disease (CKD) is both influenced by and contributes to the development of oxidative stress [216]. One of the hypothesized factors contributing to heightened oxidative stress in chronic kidney disease (CKD) is the reduced functionality of mitochondria and the subsequent increase in reactive oxygen species (ROS) production inside these organelles. In the context of diabetic kidney disease, there exists a correlation between mitochondrial malfunction and the excessive generation of reactive oxygen species, leading to cellular harm and the advancement of the illness. Inflammatory markers, such as high-sensitivity C-reactive protein, interleukin-6, tumor necrosis factor-alpha, and fibrinogen, have been found to be related to renal illness. These chemicals have the potential to activate many signaling pathways that might induce oxidative damage. Several studies conducted on rodents, such as those by Dieter et al. [217] and Galvan et al. [218], have used biosensors to validate the increase in reactive oxygen species (ROS) in mitochondria of diabetic mice. The existence of uremic toxins among individuals with chronic kidney disease (CKD) may also contribute to the occurrence of oxidative stress. Uremic toxins elicit an inflammatory response and provoke oxidative stress through the activation of polymorphonuclear cells, as well as the interleukin-1 (IL-1), interleukin-8 (IL-8), and the innate immune system. The process of uric acid (UA) generation, which occurs during the breakdown of purines by xanthine oxidoreductase, leads to the production of superoxide, hence exacerbating oxidative stress. However, there is an increasing body of research indicating that UA possesses significant antioxidant properties within a living organism [219].

In summary, acute kidney injury (AKI) plays a significant role in the initiation and progression of chronic kidney disease (CKD). The pathophysiology of acute kidney injury (AKI) encompasses common underlying factors, including cellular death or damage, as well as inflammation. Regarding this matter, clinical research provides evidence that the solitary occurrence of acute kidney injury (AKI) can lead to the development of chronic kidney disease (CKD) [220]. According to recent research, it has been shown that renal tubular epithelial cells (TECs) have a heightened vulnerability to metabolic reprogramming in cases of acute kidney injury (AKI). This reprogramming entails a shift in the metabolic activity of TECs from fatty acid oxidation (FAO) to glycolysis, mostly due to factors such as hypoxia, mitochondrial malfunction, and disrupted nutrient-sensing pathways. Nevertheless, the findings do not provide significant insights for enhancing patient outcomes, as heightened glycolysis serves as a compensatory mechanism for generating ATP. Conversely, prolonged suppression of fatty acid oxidation (FAO) and heightened glycolysis contribute to inflammatory responses, lipid buildup, and fibrosis, thereby facilitating the progression from acute kidney injury (AKI) to chronic kidney disease (CKD). Despite significant advancements, the precise mechanism responsible for the transition from acute kidney injury (AKI) to chronic kidney disease (CKD) remains mostly elusive. Animal models that accurately reproduce the process of AKI-CKD transition and the development of effective therapies are crucial for advancing our understanding of the pathophysiology of this condition [221,222].

## 14. Key Points and Highlights

In this section, we will introduce the key points and highlights of the review, providing a concise overview of the most significant findings and insights presented in the following sections.

Epithelial transport is critical for maintaining organ function, and disruptions can lead to various pathophysiological conditions;Recent advances in understanding epithelial transport have revealed complex regulatory mechanisms with potential implications for developing new therapies;Transcellular, paracellular, and vesicular transport are the three primary mechanisms of epithelial transport, with each having a unique role;Ion transport is essential for normal tissue function, and abnormalities can lead to diseases with severe consequences for multiple organ systems;Water transport is crucial for fluid balance, and disruptions can result in conditions such as dehydration and pulmonary edema;Nutrient transport is essential for overall health, and optimizing absorption can prevent malabsorption syndromes and liver diseases;Hormonal regulation of epithelial transport involves multiple hormones, and abnormalities can lead to disorders such as diabetes insipidus and hyperaldosteronism;Genetic disorders of epithelial transport can result in compromised ion, water, and substance transport, leading to pathologies in multiple organs;Understanding the different types of diarrheas and their underlying causes is important for accurate diagnosis and treatment;Hypertension is a complex condition with multiple contributing factors, and understanding physiological processes involved can provide insights into its pathophysiology;Edema is a complex condition with different forms and can have harmful effects on tissue function;Regulation of tissue fluid dynamics and prevention of edema formation involve complex interplay between various factors;Renal diseases, including acute kidney injury and chronic kidney disease, are prevalent and can lead to serious complications.

## 15. Practical Applications and Future Lines of Research

As our understanding of epithelial transport mechanisms continues to advance, there are numerous practical applications in various fields, including medical research, clinical practice, drug development, medical interventions, improved diagnostics and biomarkers, education and awareness, fluid management, diet optimization, probiotics and digestive enzymes, lifestyle modifications, and the diagnosis and treatment of hormonal imbalances.

In medical and pharmacological research, knowledge of epithelial transport mechanisms can inform the development of targeted therapies for conditions such as hypertension, heart failure, cystic fibrosis, and inflammatory bowel disease. In clinical practice, understanding epithelial transport can be applied in diagnosing and managing conditions related to epithelial barrier function, nutrient absorption, and drug absorption and distribution. In drug development, understanding vesicular transport mechanisms can aid the development of drug delivery systems for targeted drug delivery and antiviral or antimicrobial therapies. Medical interventions that correct ion transport abnormalities can benefit patients with diseases such as cystic fibrosis and renal tubular acidosis, and improved understanding of ion transport can lead to the development of diagnostic tools or biomarkers for these conditions.

Furthermore, knowledge of water transport mechanisms can be applied in diagnosing and managing conditions such as diabetes insipidus, pulmonary edema, dehydration, and heart failure, as well as in the development of therapeutic interventions. In optimizing nutrient absorption, diet optimization, probiotics, and digestive enzymes can be used to improve gut health and nutrient transport in individuals with gastrointestinal disorders. Lifestyle modifications, such as maintaining a healthy weight and avoiding high-fat diets, can also help to prevent disruptions in nutrient transport and reduce the risk of developing liver diseases. Additionally, understanding the role of hormones in regulating epithelial transport can aid the diagnosis and treatment of hormonal imbalances.

Overall, the practical applications of understanding epithelial transport mechanisms are broad and diverse, with potential implications in various fields of research, clinical practice, drug development, diagnostics, and patient care.

## 16. Conclusions

In conclusion, epithelial transport is a critical physiological process that plays a key role in maintaining the proper functioning of various organs in the human body. Recent advances in our understanding of epithelial transport have revealed complex mechanisms involving ion channels, transporters, pumps, signaling pathways, regulatory mechanisms, and the gut microbiota. Disruptions in epithelial transport can lead to various pathophysiological conditions, such as cystic fibrosis, diarrhea, hypertension, and renal diseases. Ion transport, water transport, and nutrient transport are essential for normal physiological functions, and disruptions in these processes can result in diseases and conditions affecting multiple organ systems. Hormonal regulation of epithelial transport is also crucial, and abnormalities in hormone production or signaling can lead to disorders related to epithelial transport. Genetic disorders, such as cystic fibrosis and other syndromes, can compromise ion, water, and substance transport through cell membranes, leading to various pathologies. Understanding the different types of diarrheas and their underlying causes is important for accurate diagnosis and treatment. Hypertension and edema are complex conditions involving multiple factors, and understanding the physiological processes involved can provide insights into their pathophysiology. Renal diseases, including acute kidney injury and chronic kidney disease, are prevalent and can have severe consequences for overall health. Further research in epithelial transport and related mechanisms may lead to the development of new therapies for various disorders and conditions involving disruptions in epithelial transport.

## Figures and Tables

**Figure 1 cells-12-02455-f001:**
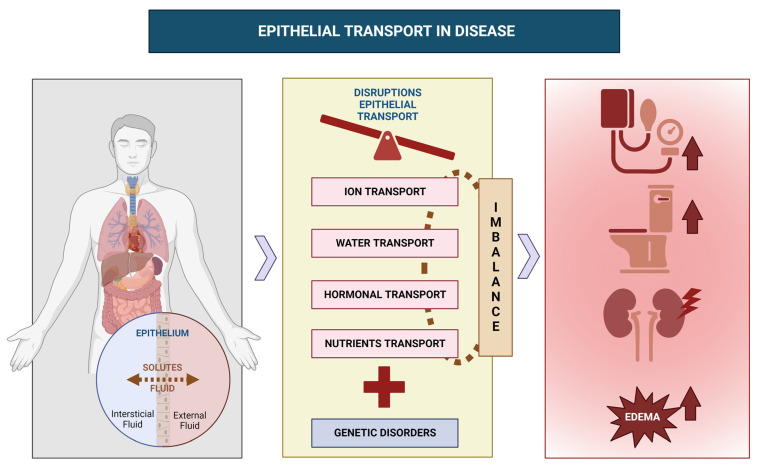
Epithelial transport in disease.

**Figure 3 cells-12-02455-f003:**
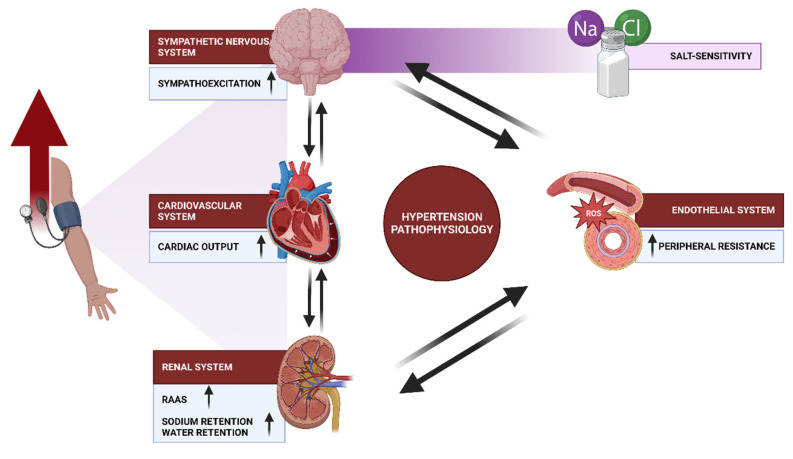
Description of the pathophysiology of hypertension and how several systems, including the sympathetic, cardiovascular, renal, and endothelial systems, are involved.

**Figure 4 cells-12-02455-f004:**
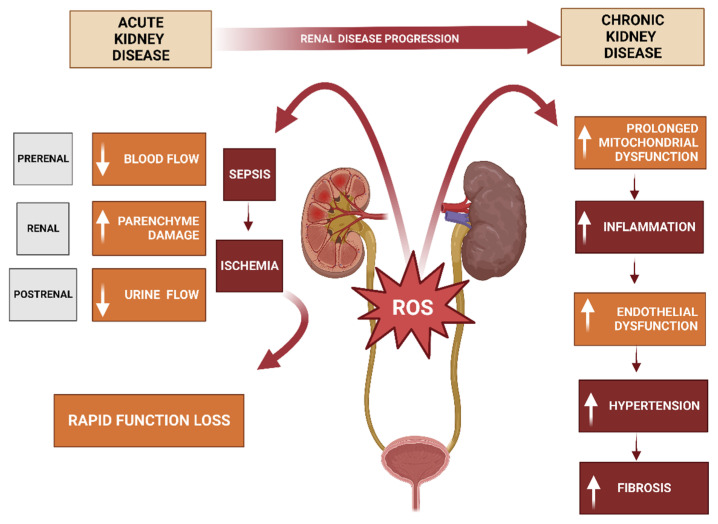
Description of the consequences that acute and chronic kidney malfunction can have on the kidneys.

**Table 1 cells-12-02455-t001:** Comprehensive Overview of Key Research Studies on Medical Conditions and Their Therapeutic Approaches.

Authors and Year	Study Title	Aim of Study	Main Outcomes	Section
Galleli et al. [170]	A Review of Its Anti-Edematous, Anti-Inflammatory, and Venotonic Properties	Discusses historical and recent pharmacological and clinical data on the anti-edematous, anti-inflammatory, and venotonic properties of escin.	Escin oral dragées and transdermal gel have both demonstrated efficacy in blunt trauma injuries and chronic venous insufficiency.	**Edema**
Pickkers et al. [200]	Acute Kidney Injury in the Critically Ill: An Updated Review on Pathophysiology and Management	Prediction and early detection of AKI, aspects of pathophysiology, and progress in the recognition of different phenotypes of AKI, as well as an update on nephrotoxicity and organ cross-talk.	Novel developments, including biomarkers and machine learning, hold promise as they are more sensitive to and predictive of the development of AKI.	**Renal disease**
Thiagarajah et al. [115]	Chloride Channel-Targeted Therapy for Secretory Diarrheas	Analyzes Enterocyte Cl^−^ channels, which represent an attractive class of targets for diarrhea therapy, as they are the final, rate-limiting step in enterotoxin-induced fluid secretion in the intestine.	Antisecretory drug therapy has considerable potential in reducing morbidity and mortality associated with infectious and some drug-induced and other diarrheas.	**Diarrhea**
Gimbrone et al. [148]	Endothelial Cell Dysfunction and the Pathobiology of Atherosclerosis	Traces the evolution of the concept of endothelial cell dysfunction, focusing on recent insights into the cellular and molecular mechanisms that underlie its pivotal roles in atherosclerotic lesion initiation and progression.	The development of pharmacomimetics of the natural, flow-mediated vasoprotective endothelial phenotype would appear to be a potentially fruitful strategy.	**Hypertension**

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
