# Peer review of "Epithelial Transport in Disease: An Overview of Pathophysiology and Treatment"

_cells, 2023, doi:10.3390/cells12202455_

Round 1

Reviewer 1 Report

Review article entitled “Pathophysiology of epithelial transport” by Vicente Javier Clemente-Suárez presented a comprehensive role of epithelial in normal and pathological conditions with some aspects of methods. By reviewing this article, I believe this article is under the scope of the “Cells” journal. Some of my suggestions here by enlisted for final decision.

1.      Diagrammatic abstract will be very interesting for this review.

2.      An abstract will be very attractive, if connectivity within the paragraphs is used.

3.      Details of literature Inclusion and exclusion criteria.

4.      Include the data regarding basolateral vesicular transport Line 125-134.

5.      Subsection:  Methods to analyze epithelial transport.

6.      Summary or map of reference research work for specific disorders like edema, renal disease etc.

Minor to Moderate revision is required

Author Response

Broad comments

Review article entitled “Pathophysiology of epithelial transport” by Vicente Javier Clemente-Suárez presented a comprehensive role of epithelial in normal and pathological conditions with some aspects of methods. By reviewing this article, I believe this article is under the scope of the “Cells” journal. Some of my suggestions here by enlisted for final decision.

Thank you for your input, we will respond to each recommendation specifically as follows:

  1. Diagrammatic abstract will be very interesting for this review.

        A graphic abstract has been included to the better elucidation in introduction

  1. An abstract will be very attractive if connectivity within the paragraphs is used.

We have rewritten the abstract ensuring smooth transitions and connections between points to enhance comprehension.

  1. Details of literature Inclusion and exclusion criteria.

        We have been more precise by adding why articles are excluded or included in this review.

  1. Include the data regarding basolateral vesicular transport Line 125-134.

       More information has been included

  1. Subsection:  Methods to analyze epithelial transport.

       Included

  1. Summary or map of reference research work for specific disorders like edema, renal disease etc.

 A table has been included at the end of the conclusions with the most relevant references depending on the disease.

Thank you for your constructive feedback; it is invaluable for refining the article.

Reviewer 2 Report

This review aims to provide an overview of the pathophysiology of epithelial transport and its role in the development of various diseases. The authors carefully gathered relevant literature to conclude the importance of the epithelial process and its regulation and suggested further research in this area.

While appreciating the topic for this review, the following comments may please be considered:

Major Comments:

·         The layout of this review is quite haphazard and makes it difficult to comprehend the actual context.

·         Although the title says ‘pathophysiology of epithelial transport’ the majority of the content discussed in this review is generalized, particularly the pathophysiology of various disorders.

·         Line 88-91; the authors say ‘recent studies’ however the references are 20 year old

·         In the methodology section line 108 says 6 reviewers and line 110 says the same team of 9 reviewers!

·         Types of epithelial transport can be divided into three subheadings. The overall paragraphing is not done correctly and there are repetitions in the information provided.

·         Para-cellular transport with the mention of tight junction protein is an old reference (2001). Please review for recent.

·         Lines 170-172 are repeated information

·         In-text referencing is not uniform (should be numbered throughout, but some have authors and number both – line 177)

·         Information regarding mitochondrial transfer, should it be in types of epithelial transport segment?

·         CFTR and related mucous thickening in the respiratory tract is mentioned more than four times. Please avoid repetition.

·         NAFLD is also repeated (431 and then 461). If there is new context, mention it in the same paragraph.

·         Pathophysiology of hypertension, edema, and renal diseases are mentioned in detail, and there is little context for epithelial dysfunction.

·         Majority of the references are more than 5 years old and quite a few are recent.

Minor comments

·         Reference 4 (line 68) investigated acute gastroenteritis in children so please add or replace a reference with more general research.

·         Reference 5 (line 74) is also about disturbance in the hypertension related to RAS and is not limited to the uptake issues with the renal tubules for electrolytes, please verify.

Reference 6 (line 80) please verify. It is more suitable in the previous paragraph on hypertension.

Reviewer 3 Report

1)      The abstract has to be re-written. The authors used plenty of “we” in the abstract. The main results obtained by previous research should be mentioned as a past participle.

2)      Methodology (line 100) is meaningless in review articles. The authors can write their aim of the review at the end of the introduction without any sub-headings.

3)      The authors wrote the authors’ contribution in line 106-112. This part should be moved to the end of the manuscript.

4)      In section 3, various types of epithelial transport should be sub-sectioned as 3.1, 3.2, and 3.3 for vascular transport, and mitochondrial transport…for better understanding.

5)      The authors did not use any figures for sections 3-9. It is easier to transfer knowledge to the audience by using of figure. Also, I would recommend adding a graphical abstract at the end of the introduction.  

Reviewer 4 Report

This is a interesting review article about the role of epithelial transport in health and disease.  The main problem I had with this review was that it was not well organized and tried to cover too much material.  I feel that the review needs to focus in more depth on a few specific diseases that are caused by problems with epithelial transport.  Instead, the review discusses the the pathophysiology of diarrhea, hypertension, edema, and renal disease, each of which are broad topics that have many causes.

Specific comments:

Introduction: Omit lines 30-48. Start at line 49.

Line 52-53.  Not sure that the transport of oxygen and carbon dioxide across the alveolar-capillary wall is selective.

lines 186-258. There is an abrupt transion to mitochondrial transfer.  What does mitochondrial transfer have to do with epithelial transport?  I'm not sure if it should be included in the review.

4. Ion transport

lines 304-331.  I would include this information in section 7 "Genetic disorders of epithelial transport.

Sections 8,9,10,11.  Too broad.  Needs to be more focused.  See previous comments.

Some parts of the review were well written.  Other parts need some editing, especially sections 6 and 7.

Round 2

Reviewer 2 Report

Line 103: Please change 'pretends' to 'intends'

Line 345-346: not clear. Please re-write

Author Response

Thank you for your valuable feedback.

  1. Line 103: I have changed "pretends" to "intends" as suggested.
  2. Line 345-346: I have revised the mentioned lines for clarity.

Thank you for bringing these to my attention. I appreciate your guidance in improving the manuscript.

Reviewer 3 Report

The authors answered my questions and performed the corrections in their paper and now I recommend publishing. 

Author Response

Dear Reviewer,

Thank you for your constructive feedback and subsequent recommendation for publication. We greatly appreciate the time and effort you dedicated to reviewing our manuscript. Your insights were invaluable in enhancing the quality of our work.

We are pleased to know that the revisions met your expectations, and we are grateful for your positive recommendation.

Thank you once again for your support and guidance.

Reviewer 4 Report

General comments:

The authors have addressed most of my main concerns.  They have improved the abstract and added new information about the role of epithelial transport in section 11. Pathophysiology of hypertension, section 12. Pathophysiology of edema, and section 13. Pathophysiology of renal disease.  I still felt that the review tries to cover an excessively broad topic.

Moderate editing of the English language is required.  For example, in several places,  such as in line 103, the word "pretends" is used incorrectly.  The abstract and introduction also need to be carefully edited for grammatical mistakes.

Author Response

Thank you for your continued feedback and for acknowledging the improvements we have made in the manuscript, especially in the sections you highlighted. We value your insights and have taken them into serious consideration.

Regarding your concern about the breadth of the topic, we understand that the review covers a wide range of subjects. Our intention was to provide a comprehensive overview, but we recognize the potential challenges this might pose for clarity and focus. We will consider narrowing down the scope or further segmenting the content in future revisions or publications to ensure depth without compromising clarity.

On the note of language quality, we sincerely apologize for the oversight. We will undertake a thorough review of the entire manuscript, with a particular focus on the abstract, introduction, and the specific lines you mentioned. We will ensure that the language is polished and that errors, such as the incorrect use of "pretends", are rectified.

In the introduction, once revised we have changed:  repetitive transitions like "subsequently" to improve flow, spelling errors such as "edoema" to "edema" and streamlined some sentences for clarity and coherence.